

**A comprehensive study of new hybrid models for Adaptive Neuro-Fuzzy**
**Inference System (ANFIS) with Invasive Weed Optimization (IWO),**
**Differential Evolution (DE), Firefly (FA), Particle Swarm Optimization (PSO)**
**and Bees (BA) algorithms for spatial prediction of groundwater spring**
**potential mapping**
Khabat Khosravi[1], Mahdi Panahi*[2], Dieu Tien Bui*[3]
1-Department of watershed management engineering, Faculty of Natural Resources, Sari Agricultural Science and
Natural Resources University, Sari, Iran.
2- Department of Geophysics, Young Researchers and Elites Club, North Tehran Branch, Islamic Azad University,
Tehran, Iran. (E-mail: panahi2012@yahoo.com)
3- Geographic Information System Group, Department of Business and IT, University College of Southeast
Norway,Gullbringvengen 36, 3800 Bø i Telemark, Norway. (E-mail: Dieu.T.Bui@usn.no)
**Abstract**
Groundwater are one of the most valuable natural resources in the world and their sustainable
management is necessary. One of the most important methods in managing groundwater is
developing groundwater potential mapping (GPM). The current study benefits from a new hybrids
of Adaptive Neuro-Fuzzy Inference System (ANFIS) with five meta-heuristic algorithms, namely
Invasive Weed Optimization (IWO), Differential Evolution (DE), Firefly (FA), Particle Swarm
Optimization (PSO) and Bees (BA) algorithms for spatial prediction of groundwater spring
potential mapping at Koohdasht-Nourabad plain, Lorestan province, Iran. A total number of 2463
springs were identified and then divided in two classes randomly, including 70% (1725 locations)
of the springs were applied for model training and the remaining 30% (738 spring locations), which
were excluded in the training phase, were utilized for the model valuation. Thirteen groundwater
occurrence conditioning factors, namely slope degree, slope aspect, altitude, curvature, stream
power index (SPI), topographic wetness index (TWI), terrain roughness index (TRI), distance from
fault, distance from river, land-use, rainfall, soil order and lithology (units) have been selected for
modeling. The stepwise assessment ratio analysis (SWARA) method was applied to determine the
spatial correlation between springs and conditioning factors. The accuracy of the map achieved
after applying these five hybrid models was determined using the area under the receiver operating
characteristic (ROC) curve (AUC). The results showed that ANFIS-DE has the highest prediction
capability (0.875) for groundwater spring potential mapping in the study area, followed by ANFIS-
IWO and ANFIS-FA (0.873), ANFIS-PSO (0.865) and ANFIS-BA (0.839). Results of Freidman
and Wilcoxon signed rank test revealed that there were statistically significant differences between
the models' performances except for ANFIS-FA vs. ANFIS-DE and ANFIS-PSO vs. ANFIS-DE.
The results of this research can be useful for decision makers to sustainable management of
groundwater resources.
Key words: Groundwater spring, ANFIS-DE, ANFIS-IWO, ANFIS-FA, ANFIS-PSO, ANFIS-
BA, Iran.
**1. Introduction**



Groundwater is defined as the water in a saturated zone which fills rock and pore spaces (Berhanu et al., 2014; Fitts, 2002) and groundwater potential is the possibility of groundwater occurrence in an area (Jha et al., 2010). The occurrence of groundwater in an aquifer is affected by various geo-environmental factors including lithology, topography, geology, fault and fracture and its connectivity, drainage pattern and land-use (Mukherjee, 1996). As one of the major conditioning factors, geological strata acts like a conduit and reservoir for groundwater. Storage and transmissivity of the formation has influence on the suitability of exploitation of groundwater in a given geological formation. Downhill and depression slopes impart runoff and improve recharge and infiltration, respectively (Waikar and Nilawar, 2014).

Groundwater which serves as a major source of drinking water to communities, for agricultural and for industrial purposes, is one of the most precious natural resources in the world (David Keith Todd and Mays, 1980) due to its consistent temperature, widespread availability, low vulnerability to pollution, low development cost and drought dependability (Jha et al., 2007). The life of about 1.5 billion people depends upon groundwater in the world solely for drinking purposes and about 38% of the irrigated lands depend on the groundwater itself (Siebert et al., 2013). As the population of mankind on earth increases, the demand for water constantly increases. The major challenge is sustainable management of groundwater to preserve and ensure continuous supply with regards to water demand. One of the most important measures for groundwater resource management is having adequate knowledge on spatial and temporal distribution of groundwater, its quantity as well as its quality. Approximately, two-third of Iran's area is covered by deserts. As a result, similar to other arid regions, the main sources of water supply for various uses, especially drinking, depends on the groundwater (Nosrati and Van Den Eeckhaut, 2012). Agriculture is one of the most prominent economic sectors in Iran and especially in the study area, limited by water scarcity (Zehtabian et al., 2010). Groundwater in Iran supplies around 65% of the water use-up and the remaining 35% is supplied by surface water (Rahmati et al., 2016). One of the most important measures to responsible for increase in fresh-water demand for drinking and agriculture is the identification of groundwater potential zoning as an essential tool for performing a successful groundwater determination, protection, and management program (Ozdemir, 2011a).

There are a number of methods for groundwater exploitation in traditional approaches including drilling as well as geological, geophysical, and hydrogeological methods. Yet, they are not only time-consuming and costly but uneconomical (David Keith Todd and Mays, 1980; Israil et al., 2006; Jha et al., 2010; Sander et al., 1996; Singh and Prakash, 2002). Recently, the application of geographic information systems (GIS) and remote sensing (RS) has become an effective procedure to groundwater potential mapping (Fashae et al., 2014) due to their ability in handling huge amount of spatial data, their easy performance and their applicability for being used in a lot of fields including water resources management, In more recent years, some probabilistic models such as frequency ratio (Oh et al., 2011), multi-criteria decision analysis (MCDA) (Kaliraj et al., 2014) (Rahmati et al., 2015) weights-of-evidence (WofE) (Pourtaghi and Pourghasemi, 2014), logistic regression (LR) (Ozdemir, 2011b; Pourtaghi and Pourghasemi, 2014), evidential belief function (EBF) (Nampak et al., 2014; Pourghasemi and Beheshtirad, 2015), decision tree (DT) (Chenini and Mammou, 2010), artificial neural network model (ANN) (Lee et al., 2012), and Shannon's entropy (Naghibi et al., 2015) have been used for recognition of groundwater potential mapping. The bivariate and multivariate statistical models have some disadvantages in measuring the relationship between groundwater occurrence and conditioning factors for the definition of statistical assumptions prior to the study (Tehrany et al., 2013; Umar et al., 2014). MCDA





technique is source of bias due to expert opinion. Traditional modeling approaches are also based on linear or additive modeling that is not consistent with natural process in the environment (Clapcott et al., 2013) but, machine learning models with non-linear structure handle data from various measurement scales and make no statistical assumptions; hence being useful for modeling applications such as GPM. ANN model is the most widely used model for environmental modeling among other machine learning models due to its computational efficiency (Bui et al., 2016; Ghalkhani et al., 2013; Rezaeianzadeh et al., 2014). The ANN model has a number of weaknesses such as poor prediction and error in modeling process (Bui et al., 2016). Thus, ANN model ensembles with fuzzy logic model and Adaptive Neuro-Fuzzy Inference System (ANFIS) model, which is a hybrid model proposed and used by some researchers due to its high accuracy (Güçlü and Şen, 2016; Lohani et al., 2012; Shu and Ouarda, 2008) (Chang and Tsai, 2016). It should be noted that even though ANFIS model has a higher accuracy than the two other model individually (Mukerji et al., 2009; Nayak et al., 2005), it has some disadvantages since it is weak in finding the best weight parameters affecting the prediction accuracy (Bui et al., 2016). These weights can be recognized using soft computing optimization process. Optimization problem is the problem of finding the best solution from among a set of all possible solutions.

The main aim of the current study is to carry out groundwater spring potential mapping (GSPM) in Koohdasht-Nourabad plain, Iran using ANFIS model, with some new metaheuristic models, namely Invasive Weed Optimization, Differential Evolution, Firefly, Particle Swarm Optimization and Bees algorithm which have some ensembles with ANFIS to solve the weakness of the ANFIS model. Another goal of the present study is drawing a comparison between prediction capabilities of these five new hybrid models in groundwater potential modeling in the study area as well. The main difference between the current study and the literature review is that these new hybrid models have not been used before for groundwater potential mapping, but their accuracy in prediction of landslide (Chen et al., 2017a) and flood (Termeh et al., 2018) susceptibility mapping has been confirmed recently. Since no such studies have been published so far in the study area, the current study is the pioneer work in this subject.

## 2. Case study description

Koohdasht-Nourabad Plain is located in the west part of Lorestan province, Iran. It lies between 33°3′ 28 and 34° 22′ 55 N latitudes and 46° 50′ 19 to 48° 21′ 18 E longitudes (Fig. 1). The region is located in the semi-arid area with mean annual precipitation of about 450 mm. The plain covers around 9531.9 km$^2$ with the population of 362,000 people (according to 2016 census). The occupation of most people living in the region is farming and water requirements are met through groundwater extraction. The altitude of the study area varies between 531 m and 3175 m above the sea level while the maximum and minimum slope is 0$^o$ and 64$^o$, respectively. Geologically, the study area is located in Zagros structural zone of Iran and is mostly covered by Quaternary and Cretaceous-Paleocene geologic time scale. The dominant land-use of the study area is moderate forest (20%) and rocks covers the smallest area percentage (0.00067%). The residential areas also covers about 3% of the Koohdasht-Nourabad plain. Rock crop/Inceptisoils are the dominant soil types in the study area, covering about 51% of the study area.





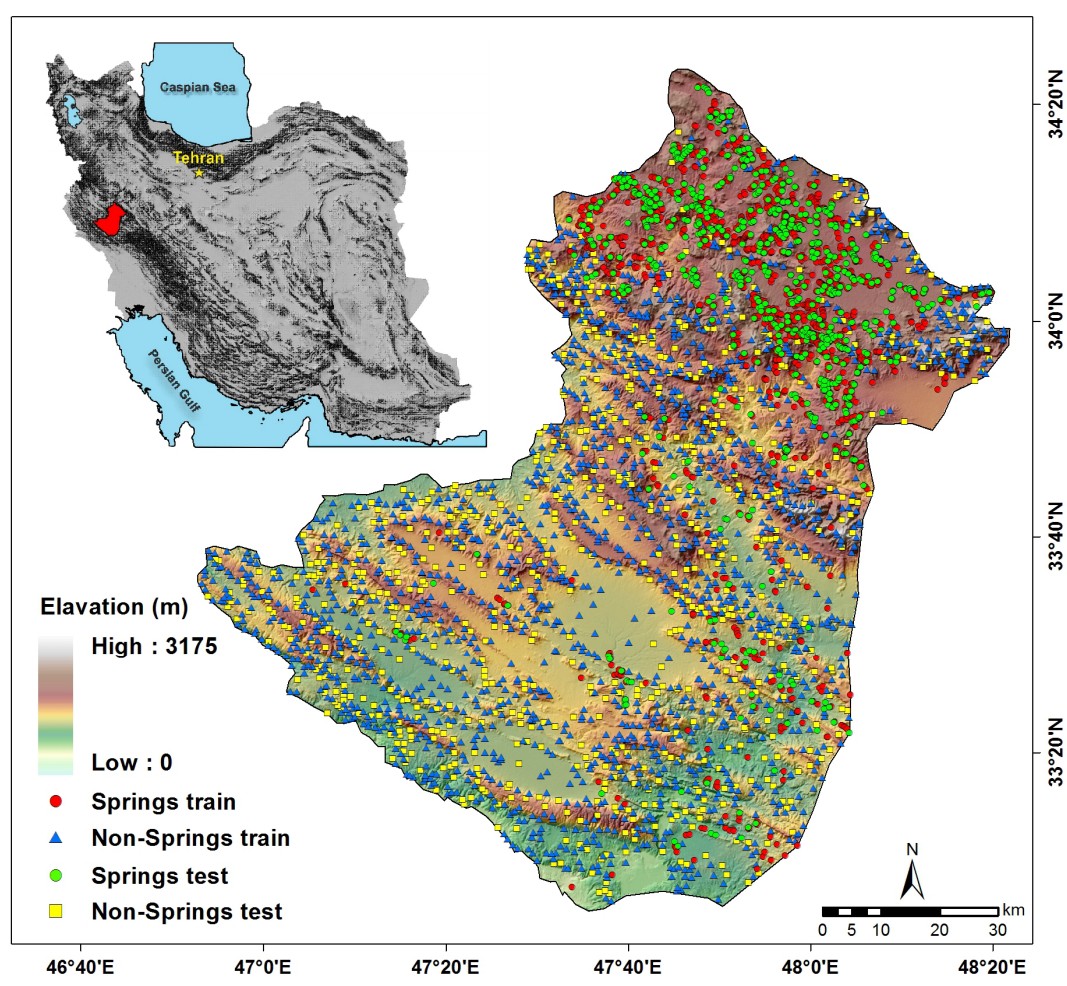

Fig.1. Groundwater well locations with DEM of the study area

## 3. Methodology

The methodological approach has been shown in Fig 2 and will be described step by step below.





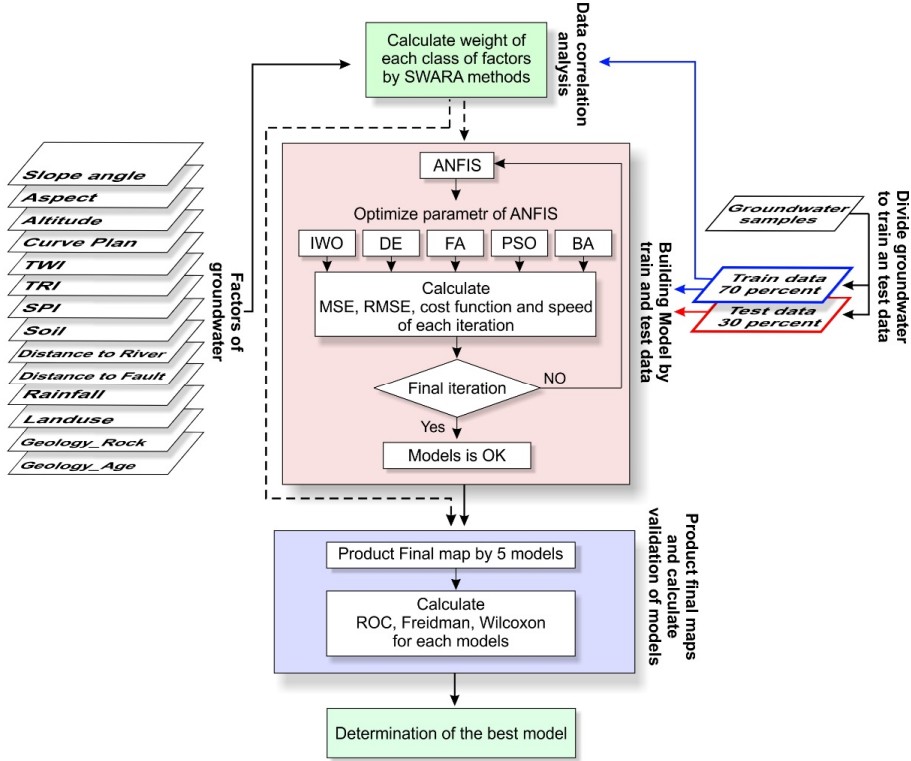

Fig.2. Conceptual model adopted in the current study

### 3.1. Data preparation

#### 3.1.1. Groundwater spring inventory map

In any spatial prediction modeling such as groundwater modeling, spatial relationship between occurrence of groundwater springs and the conditioning factors should be analyzed. In Koohdasht-Nourabad plain, a total of 2463 springs were selected and considered for modeling. Most of the spring locations were checked using extensive field survey with GPS hand hole.

#### 3.1.2. Construction of the training and validation dataset

Spatial prediction of groundwater potential mapping, using machine learning model, is a binary classification because the groundwater potential index is divided into two parts including spring location and non-spring location. Thus, 2463 non-spring locations were constructed randomly using created random point command in ArcGIS10.2. According to Chung and Fabbri (Chung and Fabbri, 2003), it is impossible to validate the model performance without considering the dataset for the two parts (Chung and Fabbri, 2003). The first part is used for model building which is also called training dataset and the other part is utilized for validating or testing the model performance which also called testing dataset (Pham et al., 2017a). In this study, a ratio of 70/30 was selected randomly for training and testing the dataset (Pourghasemi et al., 2013a; Pourghasemi et al., 2012; Pourghasemi et al., 2013b; Xu et al., 2012). Both spring location and non-spring location have





been divided into two groups for training (1725 location) and validating (738 location) purposes (Fig 1).

For building the training dataset, a total number of 1725 locations were randomly selected for both spring and non-spring location and were then combined with each other, and 738 of the remaining location of springs and non-springs were combined with each other again to construct the testing dataset. Finally, both training and testing datasets were converted to raster format and then overlaid with 13 groundwater conditioning factors. In both training and testing dataset, the spring pixels were assigned a value of 1 and non-spring pixels were assigned 0 (Bui et al., 2015).

### 3.1.3. Groundwater conditioning factor analysis

#### 3.1.3.1. Selection of the Groundwater conditioning factor and multi-collinearity analysis

After definition of the effective factors, the conditioning factor should be assessed for multi-collinearity problem. Multi-collinearity takes place when two or more non-independence conditioning factors are highly correlated, or in other words inter-dependent (Li et al., 2010). Several methods have been proposed to multi-collinearity diagnoses from among which, two methods of Variance Inflation Factor (VIF) and tolerance are widely used for multi-collinearity problem recognition (Bui et al., 2016; O'brien, 2007). The VIF greater than 5 and tolerance less than 0.1 shows the multi-collinearity problem (Bui et al., 2011; O'brien, 2007).

In the current study, 14 conditioning factors have been selected including slope degree, slope aspect, altitude, curvature, stream power index (SPI), topographic wetness index (TWI), Terrain roughness index (TRI), distance from fault, distance from river, land-use, rainfall, soil order and lithology units. These factors have been selected according to the literature review, characteristics of the study area, and data availability (Mukherjee, 1996; Nampak et al., 2014; Oh et al., 2011; Ozdemir, 2011a), but there isn't any agreement on which the factors to be used for modeling. The process of converting continuous variables into categorical classes were carried out using expert opinions in order to define the class intervals (Bui et al., 2011). Digital Elevation Model (DEM) has been downloaded from ASTER global DEM with 30x30 m grid size and then slope degree, slope aspect, altitude, curvature, SPI, TWI and TRI have been constructed using DEM. Slope degree of the study areas varies between 0-64 degree. Slope factor has a direct impact on the runoff generation and therefore groundwater recharge, as the lower the slope, the lower runoff generation and the higher groundwater recharge. The slope degree has been divided in five categories using quantile classification scheme (Tehrany et al., 2013; Tehrany et al., 2014), including 0-5.5, 5.5-12.11, 12.11-19.4, 19.4-28.7, 28.7-64.3 degree (Fig 3a). Slope aspect is another factor that has affects the groundwater potential through solar radiation since the north aspect receives a lower sun light and as a result is less wet or has low evapotranspiration. The slope aspect has been provided in 5 different classes including, flat, north, west, south and east (Fig 3b). The third conditioning factor is altitude. It has been divided into five classes using quantile classification scheme including 531-1070, 1070-1385, 1385-1703, 1703-2068 and 2068-3175 m (Fig.3c). Curvature factor has been constructed using DEM and finally divided into three classes, namely concave (<−0.05), flat (−0.05–0.05), and convex (>0.05) (Fig.3d) (Pham et al.2017). SPI is the measurement of erosive power of surface runoff and TWI shows an amount of the flow that accumulates at any point in the catchment. TRI, topographic roughness or terrain ruggedness calculates the sum of change in elevation between a grid cell and its neighborhood. SPI, TWI and TRI have been constructed in the system for Automated Geoscientific Analyses (SAGA-GIS 2.2)



software and finally divided into five classes that are 0-48664, 48664-227099, 227099-583969,
583969-1330153, 1330153-4136452 (Fig.3e) for SPI, 2.1-4.6, 4.6-5.6, 5.6-6.6, 6.6-7.9, 7.9-11.9
(Fig.3f) for TWI, and finally 0-8.7, 8.7-18.2, 18.2-29.9, 29.9-46.6, 46.6-185 (Fig.3g) for TRI.
Distance from fault and river factors have been provided using fault and river of the study area via
multiple ring-buffer command in ArcGIS10.2 software which is finally divided into five classes
including: 0-200, 200-500, 500-1000, 1000-2000 and >2000 m (Fig. 3h and Fig. 3i). Lithology
plays a key role in determining the groundwater potential occurrences due to different infiltration
rate of formation that has been considered in some previous studies (Adiat et al., 2012; Nampak et
al., 2014; Pradhan, 2009). Land-use of the study area has been provided through Landsat 7
Enhanced Thematic Mapper plus (ETM+) images downloaded from the US Geological Survey
(USGS) and supervised image classification techniques (Lillesand et al., 2014). Finally, the
accuracy of the land-use map has been controlled by filed surveys.
Twenty five land-use types including agriculture, garden, dense-forest, good rangeland, poor
forest, waterway, mixture of garden and agriculture, mixture of agriculture with dry farming,
mixture of agriculture with poor-garden, dry farming, follow, dense rangeland, very poor forest,
mixture of waterway and vegetation, mixture of moderate forest and agriculture, mixture of
moderate rangeland and agriculture, mixture of poor rangeland and follow, mixture of low forest
and follow, wood-land, moderate forest, moderate rangeland, poor rangeland, bare soil and rock,
urban and residential, mixture of very poor forest, and rangeland have been identified (Fig.3j). As
the major source of recharge to the groundwater, rainfall has been provided via mean annual
historical rainfall data of past 15 years (2000–2015) using 4 rain-gauge stations in the study area.
Inverse distance weighted (IDW) method has been used for the preparation of the rainfall map due
to lower RMSE than other methods and then, rainfall map of the study area has been divided into
five categories including: 300-400, 400-500, 500-600, 600-700, 700-800 mm (Fig 3k). The soil
properties directly affect the water infiltration rate as well as groundwater recharge. The 1:50000
soil map of Lorestan province obtained from the Iranian Water Resources Department (IWRD)
has been used for the analysis. The soil map was in a polygon format which needed to be converted
to grid. The most dominant feature of the study area is rock outcrop/Entisols, rock
outcrop/Inceptisols, Inceptisols, Inceptisols/Vertisols and Badlands (Fig.3l).





Fig.3. Thematic Groundwater conditioning factor in the study area: slope degree(a), slope aspect (b),
altitude (c), curvature (d), SPI (e), TWI (f), TRI (g), distance from fault (h), distance from river (i), land-
use (j), rainfall (k), soil order (l), and lithology units (m).




Fig.3.Continued





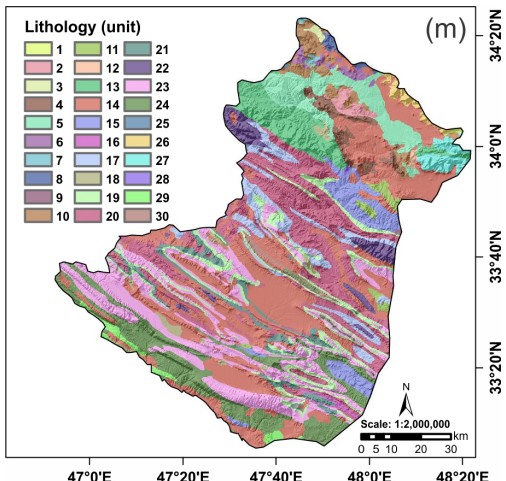


Fig.3. Continued

Finally, all the aforementioned groundwater conditioning factors for modeling purposes were
converted to a raster grid with 30 m × 30 m pixel size in the ArcGIS 10.2 software. Lithology
(unit) has a high influence on infiltration; thus, it has been considered in the current study.
Lithology for the study area has been constructed in scale of 1:100000, which was created by
Iranian Department of Geology Survey (IDGS) and divided into thirty classes including: OMq,
PeEf, PlQc, K1bl, Plc, pd, TRKubl, TRJvm, MPlfgp, OMql, Plbk, E2c, TRKurl, Qft2, MuPlaj,
KEpd-gu, Kgu, Qft1, Ekn, KPeam, PeEtz, Kbgp, EMas-sb, Mgs, TRJlr, Klsol, JKbl, Kur, OMas
and Mmn (Fig.3m).
**3.2. Spatial relationship between spring location and conditioning factors**
Step-wise Assessment Ratio Analysis (SWARA) as a Multi-Criteria Decision Making (MCDM)
was first introduced by Keršuliene in 2010 for the first time (Keršuliene et al., 2010) as a Multi-
Criteria Decision Making (MCDM). Since this method is both simple and rooted on experts'
views, it has drawn a lot of attention in diverse fields (Alimardani et al., 2013; Hong et al., 2017).
The specialist allocates respectively the highest and lowest rank from the most and least valuable
criterion, respectively. Afterwards, the all-inclusive ranks are specified by the average value of
ranks. SWARA's detail is illustrated in Fig 4.





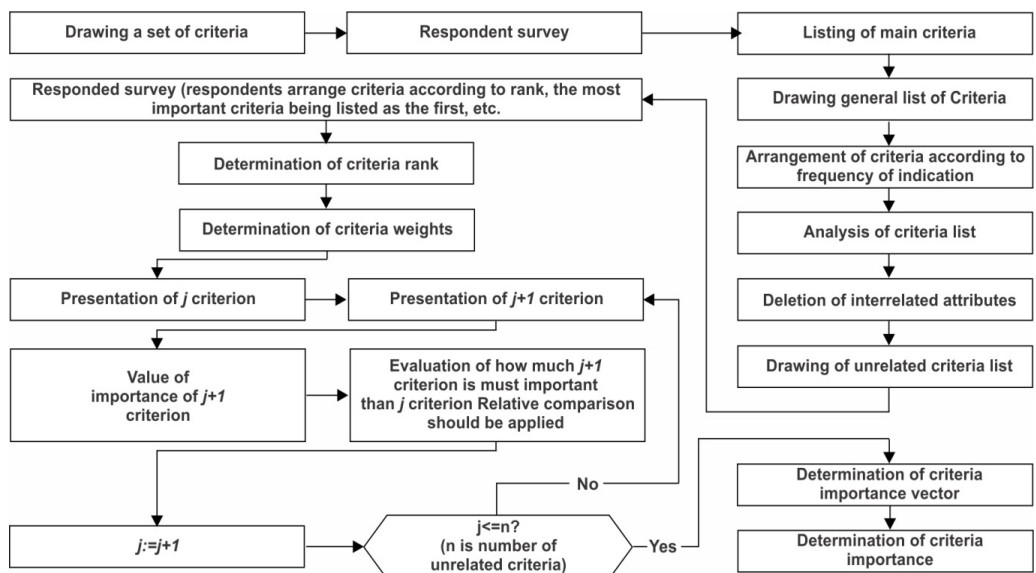

254      Fig4. the flowchart of SWARA method (Keršuliene et al., 2010)

255 The phases of method are as the following:

256 Phase one: for evolving decision making models, first, the experts define the problem solving
257 criteria. By using the practical knowledge of the experts, the priority for each criteria are
258 determined as well and the criteria are organized in descending order finally.

259 Phase two: regarding to each parameter's ranking, the following trend is employed for calculation
260 of the weight in each criteria:

261 Starting from the second criterion, the respondent explains the relative importance of the criterion
262 $j$ in relation to the $(j-1)$ criterion, and for each particular criterion as well. As Keršuliene
263 mentioned in his article, this process specifies the Comparative Importance of the Average
264 Value, $S_j$ as follows (Keršuliene et al., 2010):

265 $$S_j = \frac{\sum_i^n A_i}{n} \qquad (1)$$

266 Where $n$ is the number of experts; $A_i$ explicates the offered ranks for each factor by the experts; j
267 stands for the number of the factor.

268 Subsequently, the coefficient $K_j$ is determined as follows:

269 $$K_j = \begin{cases} 1 & j = 1 \\ S_j + 1 & j > 1 \end{cases} \qquad (2)$$

270 Recalculation of weight$Q_j$ is as the following:




$$Q_j = \frac{X_{j-1}}{K_j} \tag{3}$$
The relative weights of the evaluation criteria are calculated by the following equation:
$$W_j = \frac{Q_j}{\sum_{j=1}^{m} Q_j} \tag{4}$$
Where $W_j$ shows the relative weight of j-th criterion, and m stands for the total criteria number.

### 3.3. Groundwater spring prediction modelling

In this research, five new hybrid models namely ANFIS-DE, ANFIS-IWO, ANFIS-FA, ANFIS-
PSO, ANFIS-BA were utilized for the analysis of determination of groundwater potential zonation
in the study areas and for comparison between their prediction capabilities.

### 3.3.1. Adaptive Neuro-Fuzzy Inference System

Adaptive Neuro-Fuzzy Inference System (ANFIS) is obtained from the combination of Artificial
Neural Network (ANN) and fuzzy logic (Jang, 1993). ANFIS is more efficient than the two
mentioned models. Therefore, ANN has the automatic ability but is not able to explain how to get
the output from decision making. Fuzzy logic, on the other hand, is the reverse of ANN by
generating output from fuzzy logical decision without the ability of self-operating learning
(Aghdam et al., 2017; Chen et al., 2017b; Phootrakornchai and Jiriwibhakorn, 2015). ANFIS was
proposed by Jang in 1993 (Jang, 1993) to solve nonlinear and complex problems in one framework
(Rezakazemi et al., 2017). This model has been used in date processing, fuzzy control and others
fields (Zengqiang et al., 2008). The members of ANFIS are the function parameters from dataset
for describing the system behavior (Jang, 1993). ANFIS applies to Takgi-Sugeno-Kang (TSK)
fuzzy model with two rules of "If-Then" with two inputs $x_1$ and $x_2$, and one output $f$ (Takagi and
Sugeno, 1985), as follows:
$$Rule2\ 1: if\ x_1\ is\ A_1\ and\ x_2\ is\ B_1, then\ f_1 = p_1x_1 + q_1x_2 + r_1 \tag{5}$$
$$Rule\ 1: if\ x_2\ is\ A_2\ and\ x_2\ is\ B_2, then\ f_2 = p_2x_2 + q_2x_2 + r_2 \tag{6}$$
Jang's ANFIS consists of feed-forward neural network with six distinct layers. The ANFIS
architecture is shown in Fig. 5. There are two shapes for nodes by different concepts in Fig.4.
Fixed nodes are represented by the circular nodes and square nodes are adaptive nodes.





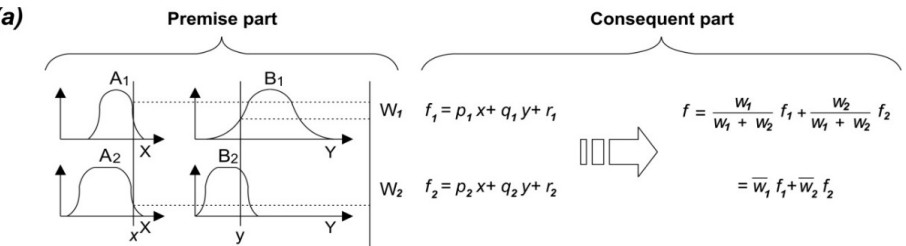

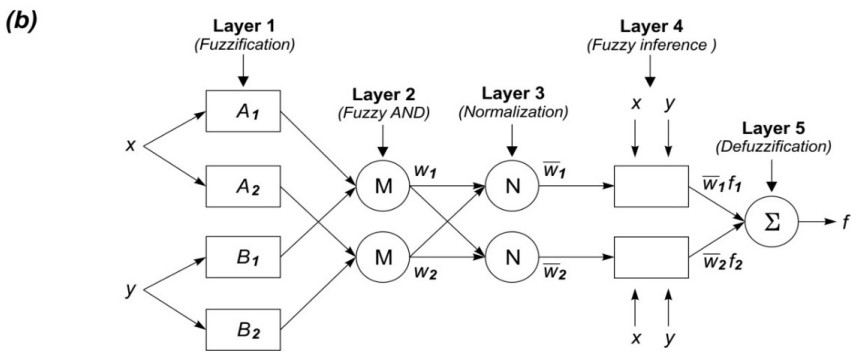

Fig.5. General ANFIS architecture of first order Takagi-Sugeno Fuzzy Model (Jang, 1993).

Layer 1 (input layer): This layer is input layer of $x_1$, $x_2$.

Layer 2 (fuzzification layer): Fuzzification layer converts input variables into fuzzy values and input is normalized. The parameter of i is an adaptive node with a function of $Q_{1,i}$ to compute full, none or partial membership.

$$Q_{1,i} = \mu_{A_i}(x_1) \ for \ i = 1,2 \tag{7}$$

$$Q_{1,i} = \mu_{B_{i-2}}(x_2) \ for \ i = 3,4 \tag{8}$$

Layer 3 (antecedent layer): This layer has two fixed nodes by the symbol of $\pi$ to compute firing strength ($w_i$) or ($Q_{2,i}$) of every rule where outputs of second layer are multiplied.

$$Q_{2,i} = w_i = \mu_{A_i}(x) \times \mu_{B_i}(y) \ for \ i = 1,2 \tag{9}$$

Layer 4 (strength normalization layer): Fixed nodes of this layer are shown by the word of "N". These nodes apply to calculating the ratio of individual rule's firing strength to the sum of all rules' firing strengths. Output of this layer is called normalized firing strength ($Q_{3,i}$).

$$Q_{3,i} = \frac{w_i}{\sum w_i} = \frac{w_i}{w_1 + w_2} = w_i \ for \ i = 1,2 \tag{10}$$

Layer 5 (consequent layer): This layer is known as defuzzification layer and determines a function for each adaptive square node.





$$Q_{4,1} = w_i.f_i = w_i.(p_i x + q_i y + r_i) \quad for \ i = 1,2 \tag{11}$$


Where $p_i$, $q_i$ and $r_i$ are the consequent parameters.
Layer 6 (inference layer): Result of this layer is the overall output which is computed as the sum
of all incoming signals from the defuzzification layer by the label of "$\sum$" in fig.5.

$$Q_{5,1} = \sum w_i.f_i = \frac{\sum w_i.f_i}{\sum w_i} = f_{out} \tag{12}$$


### 321    3.3.2. Meta-heuristic optimization

The main goal of this phase is to find the optimal antecedent and the consequent parameters of
the ANFIS model using IWO, DE, FA, PSO, and Bee algorithms.

### 325    3.3.2.1. IWO algorithm

Invasive weed optimization is one of the metaheuristic algorithms which mimics the colonizing
behavior of weeds. Its design is based on the way to find proper place for growth and reproduction
of weeds by Mehrabian and Locus (Mehrabian and Lucas, 2006). One characteristic of this
algorithm is its simplified structure; the number of input parameters is low and has strong
robustness. Furthermore, it is easy to understand and the same merit causes it to be used for solving
difficult nonlinear optimization problems (Ghasemi et al., 2014; Naidu and Ojha, 2015; Zhou et
al., 2015). Moreover, by comparing the results of IWO algorithm and other algorithms like SFLA
and PSO for solving optimization problems, IWO algorithm can compete with other ones
(Ghasemi et al., 2014). This algorithm consists of 4 parts as following:
1- Initialization
Random spread of some limited weeds in searching area with dimension D is considered as the
initial population of solutions.
2- Reproduction
Weeds are able to reproduce some seeds in accordance with their fitness during their growth. In
other words, the number of produced seeds from $S_{min}$ value for weeds starts with Worst fitness
and then increases in linear fashion to $S_{max}$ for them with best fitness (Fig. 6).



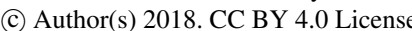


Fig.6. Seed production procedure in a colony of weeds (Mehrabian and Lucas, 2006).


3- Spatial dispersal
Produced seeds are distributed in the searching area randomly in a way that is located close to their
families with normal distribution, their mean equal to zero, and different variances. Moreover,
standard deviation decreases in each iteration from $\sigma_{min}$ to $\sigma_{max}$ and is calculated by the following
non-linear equation:
$$\sigma_{iter} = \frac{(iter_{max} - iter)^n}{(iter_{max})^n}(\sigma_{min} - \sigma_{max}) + \sigma_{max} \qquad (13)$$

Where $iter_{max}$ is the last iteration, $\sigma_{iter}$ is the standard deviation of iteration, and n is the non-
linear index considered between 2 and 3 (Saravanan et al., 2013).
4- Competitive exclusion
All weeds and their seeds combine in order to make up the population of next generation. If the
population exceeds a definite maximum, those weeds with lower fitness will be removed. The
reproduction mechanism and the competition provide breeding opportunity for proper weeds. If
they generate fitter offspring, the offspring can survive the competition.
5- Termination Condition
Step 2 to 4 repeat in order for the iteration to reach its maximum defined value and the weeds with
the best fitness will be the nearest condition to optimal solution.
**3.3.2.2. DE algorithm**
DE is another popular algorithm used as an evolutionary algorithm in recent years used for finding
global optimal answers in a problem with continuous space  (Chen et al., 2017a; Das et al., 2009).
This method was first introduced by Storn and Price (Storn and Price, 1997). It is very similar to
genetic algorithm that produces next optimum generation by three operators: mutation, crossover,
and selection. This algorithm starts by producing random population in which each individual of





population is a symbol of solution to the problem. Vector $X_i^G = (x_{1,i}^G, x_{2,i}^G, x_{3,i}^G, \dots, x_{D,i}^G)$ shows each
individual of population i = {0,1,2, ..., NP} is a number of each individual, in which D stands for
the search dimension or in other words, is a component problem and $G = \{0,1,2, \dots, G_{max}\}$
generation time that $G_{max}$ is the total number of generations. By assuming the maximum and
minimum of every dimension of searching space, there are $X_L = \{x_{1,L}, x_{2,L}, \dots, x_{D,L}\}$ and $X_U = $
$\{x_{1,U}, x_{2,U}, \dots, x_{D,U}\}$, respecitivly; initial population is defined as the following (Storn and Price,
1997):

$x_{j,i}^0 = x_{j,L} + rand(0,1).(x_{j,U} - x_{j,L})$ (14)
Where $rand(0,1)$ is a uniformly distributed random number in [0, 1]
3.3.2.2.1. Mutation
The first operator in DE algorithm is mutation, which produces mutant vector $V_i^G = $
$\left(V_i^G, V_2^G, \dots, V_D^G\right)$ by using each individual which is called target vector. Four well-known mutant
operators that are used are as the following:
DE/rand/1 : $V_i^G = X_{r1}^G + F.(X_{r2}^G - X_{r3}^G)$
DE/rand/2 : $V_i^G = X_{r1}^G + F.(X_{r2}^G - X_{r3}^G) + F.(X_{r4}^G - X_{r5}^G)$
DE/best/1 : $V_i^G = X_{best}^G + F.(X_{r1}^G - X_{r2}^G)$
DE/best/2 : $V_i^G = X_{best}^G + F.(X_{r1}^G - X_{r2}^G) + F.(X_{r3}^G - X_{r4}^G)$
DE/current − to − rand/1 : $V_i^G = X_i^G + F.(X_{r1}^G - X_i^G) + F.(X_{r2}^G - X_{r3}^G)$
DE/current − to − rand/1 : $V_i^G = X_i^G + F.(X_{best}^G - X_i^G) + F.(X_{r1}^G - X_{r2}^G)$ (15)
r1, r2, r3, r4, are the integer numbers that have been chosen randomly from [0,NP] and the
condition of $r1 \neq r2 \neq r3 \neq r4$ exists. F is the Scale factor that determines the mutation scale. It
is generally selected as a random number from [0,1]. $X_{best}^G$ is an individual that has the best fitness
value in G generation.
3.3.2.2.2. Crossover
The purpose of this step is to produce trail vector $(U_{ij})$. Thus, this operator is defined by replacing
some elements of the target vector $X_i^G$ with mutant vector $V_i^G$ as the following (Storn and Price,
1997):

$U_{ij} = \begin{cases} V_{ij}^G & if\ rand\ [0,1] \leq CR\ or\ j = j_{rand} \\ X_{ij}^G & otherwise \end{cases}$ (16)
Where i ∈ {1,2, ..., NP}, j ∈ {1,2, ..., D}, $j_{rand}$, is a random number from [1,D] and CR is the
crossover rate which is uniformly distributed random number in [0,1].
3.3.2.2.3. Selection





Selection is characterized by comparing fitness value of $U_{ij}$ trail vector with the target vector $\left(X_i^G\right)$
and choosing the best ones as the next generation (Storn and Price, 1997).
$$X_i = \begin{cases} U_i^G & if \ f(U_i^G \leq f(X_i) \\ X_i^G & otherwise \end{cases} \qquad (17)$$

### 3.3.2.3. FA algorithm

Researches always try to design powerful evolutionary algorithms by utilization of swarm social
behavior of animals, insects, and plants to use them for problem solving (Poursalehi et al., 2015).
Firefly algorithm has been defined by Yang in Cambridge University (Yang, 2009) as an
evolutionary algorithm. In recent years, many researches in different fields have taken advantage
of this algorithm for optimization. The results of using FA algorithm in different problems, which
require optimization, have been better than other algorithms such as SA, GA, PSO, and HAS
(Alweshah and Abdullah, 2015). This algorithm is known as meta-heuristic algorithm that is
originated from flashing and communication behavior of fireflies (Yang, 2009; Yang, 2010).
Somewhere in the region of 2000, special firefly species exist that most of which produce short
and rhythmic flashes (Zeng et al., 2015). Like in every other swarm intelligence algorithm, where
their components are known as solutions for the problems, in this algorithm each firefly is a
solution and its light intensity is the objective function value. In other words, a firefly with more
light intensity is known as a solution. On the other hand, this firefly attracts more fireflies.
Generally, firefly algorithm follows three idealized rules as below: 1- All firefly species are unisex,
with each of them attracting other fireflies without considering their gender (Amiri et al., 2013).
2- Attractiveness of a firefly is related to its light intensity. Thus, from two flashing firefly species,
the one with lower light intensity moves toward the other one with higher light intensity. It should
be noted that the distance between fireflies is significant because the farther they are from each
other, the dimmer the light gets and the attractiveness declines exponentially (Gandomi et al.,
2013). Moreover, if the light intensity of fireflies were the same; they would move randomly
(Senapati and Dash, 2013). 3- Light intensity of a firefly is defined as an objective function value
and must be optimized.
In order to design FA, two substantial issues are needed to be defined: light intensity variation (I)
and the attractiveness' formulation (β). Fireflies' attractiveness is determined by their light
intensity or brightness. In addition, brightness is associated with the objective function. The light
intensity I(r) varies with the distance r monotonically and exponentially as:
$$I(r) = I_0 e^{-\gamma r^2} \qquad (18)$$
where I is the original light intensity, $\gamma$ is the fixed light absorption coefficient and r is the distance
between the two fireflies. Also, attractiveness rate is defined as below:
$$\beta = \beta_0 e^{-\gamma r^2} \qquad (19)$$
where $\beta_0$ is the attractiveness when r=0. Also, the distance between two fireflies i and j with $X_j$
and $X_j$ is determined by the following equation:



$$r_{ij} = \|X_i - X_j\| = \sqrt{\sum_{k=1}^{d}(X_{i,k} - X_{i,k})^2} \qquad (20)$$
where $d$ is the number of the problem dimensions and $X_{i,k}$ is the $k-th$ element of the $i-th$
firefly. Also, the movement of a firefly $i$ which is attracted to another attractive firefly $j$, is
determined by (Yang, 2009):
$$X_i = X_i + \beta_0 e^{-\gamma r_{ij}^2}(X_j - X_i) + \alpha(rand - \frac{1}{2}) \qquad (21)$$
In Eq. (21), the first and the second terms determines the attraction. However; the third term is
regarded as a randomization with α, which is the step parameter, and ultimately, the rand is a
random number generator which is uniformly distributed in a range from 0 to 1.

### 444 3.3.2.4. PSO algorithm

As a Meta-heuristic algorithm, PSO was first designed by Eberhart and Kennedy (Eberhart and
Kennedy, 1995). Sensible characteristics of this algorithm include being powerful for optimizing
the non-linear problems, its quick convergence, and relatively low calculations. These
characteristics have made distinctions between this algorithm and other algorithms (Cheng et al.,
2010). Thus, PSO algorithm in those problems that need optimization has a special place among
researches. This algorithm has been inspired by the way the birds and fish use their collective
intelligence for finding the best way to get food (Kennedy, 2011; Kennedy and Eberhart, 1995).
Therefore, each bird implemented in this algorithm acts as a particle that is in fact a representative
of solution to problems. These particles find the optimum answers for the problem by searching in
"n" dimension space whereas "n" is the number of problem's parameters. For this purpose,
particles were scattered randomly in considered space at the beginning of algorithm
implementation. Then, the positioning in each iteration can improve by using equation 1 and 2 and
finding better situations in that iteration and the best position of particles vector addition.
Assuming that $x_i^t = (x_{i1}^t, x_{i2}^t, ..., x_{in}^t)$ and $v_i^t = (v_{i1}^t, v_{i2}^t, ..., v_{in}^t)$ are the position and velocity of the
"$i-th$" particle in "t th" iteration, respectively. Then, position and velocity of "i th" particle in
"$(i + 1)$ th" iteration is calculated by summing equation 1-2 (Eberhart and Kennedy 1995).
$$v_i^{t+1} = \omega v_i^t + c_1 r_1(p_i^t - x_i^t) + c_2 r_2(g_i^t - x_i^t) \qquad with \quad -v_{max} \le v_i^{t+1} \le v_{max}$$
$$x_i^{t+1} = x_i^t + v_i^{t+1} \qquad (22)$$
where $x_i^t$ is the last position of "i th" particle, $p_i^t$ the best found position by "i th"
particle, $g_i^t$ the best found location by particles, $r_1, r_2$ the random number between 1
and 0. $\omega, c_1$ and $c_2$ the inertia weight, cognitive coefficient, and social coefficient,
respectively. In order to value them, many papers have been presented (Olsson, 2010)
and finally the following equation has been used (Nieto et al., 2015).
$$\omega = \frac{1}{2ln2} \quad and \quad c_1 = c_2 = 0.5 + ln2 \qquad (23)$$
It is noteworthy that the algorithm continues until the best found position by each
particles unifies with the best found position of particles. In other words, all particles
accumulate in one position and actually the answer to the problem is optimized.





### 3.3.2.5. Bee algorithm

One of the meta-heuristic algorithms designed according to bee swarm-based is Bee Algorithm. This algorithm which was first introduced by Pham (Pham et al., 2005; Pham et al., 2011) is inspired by foraging behavior of bees' colonies in search of food sources (flower patches) located near the hive. In the beginning, evenly distributed scout bees are scattered randomly in different directions to identify flower patches. After that, scout bees come back to hive and start a specific dance called waggle dance. This dance is for communicating with others in order to share the information of discovered flower patches. This information indicates direction, distance, and nectar quality of the flower patches. All the information helps the colony to have proper evaluation of all flower patches. After evaluation, scout bees come back to the location of discovered flower patches with other bees named recruit bees. Regarding the distance and the amount of nectar, different number of recruit bees are assigned to each flower patch. In other words, those flower patches with better nectar quality dedicate more recruit bees to themselves. Following that, recruit bees evaluate the quality of flower patches when performing the harvest process so that they leave the flower patches if they have low quality. Conversely, if the flower patch quality is good, it will be announced during the next waggle dance. Before implementing the BA algorithm, the following parameters need to be defined:

The number of scout bees (n), the number of patches selected out of n visited points (m), the number of best patches out of m selected patches (e), the number of bees recruited for e best patches (nep), the number of bees recruited for other (m–e) selected patches (nsp), the size of patches (ngh) and the stopping criterion.

At first, "n" number of scout bees with uniform distribution is scattered in search space randomly. Then, the algorithm starts to evaluate the fitness of those seen places by scout bees in order to define and select suitable bees as elite bees.

The sites of elite bees are selected from local search and the algorithm implements the neighborhood searches within the selected bees' sites for the best ones where more bees exist. Only the proper bee is chosen to survive the next bee population in each site and other bees are allocated around the search space randomly to find new potential solutions. These steps continue until the algorithm convergences.

### 3.4. Model's performance assessment

Forecasting error as the quantitative approaches, define as the difference between the observed and estimated values which have been used for determination of the accuracy of the performed models. In the current study the model prediction capabilities for each hybrid model in terms of spatial groundwater prediction was evaluated using Mean Squared Error (MSE) as follows (Tien Bui et al, 2016):

$$MSE = \frac{\sum_{i=1}^{n}(O_i - E_i)^2}{N}$$

(24)





Where $O_i$ and $E_i$ are observation (target) and prediction (output) values in both training
and testing dataset and N is the total samples in the training or the testing dataset.

**3.5. Model's performance validation and comparisons**

According to Chung and Fabbari (Chung and Fabbri, 2003), validation is one of the
most important steps in any spatial prediction modeling and without validation, the
result of the models do not have any scientific significance. Prediction capability of
these five spatial groundwater models must be evaluated using both success-rate and
prediction-rate curves (Hong et al., 2015). Success-rate curves show how suitable the
built model is for the groundwater potential assessment or for the evaluation of the
goodness of fit (Gaprindashvili et al., 2014). Success-rate curves have been constructed
using groundwater potential maps and the number of spring locations used in training
dataset (Pradhan et al. 2010). Prediction rate curves which show the probabilities of the
groundwater occurrences demonstrate how good the model is or evaluate the prediction
power of the models. Therefore, it can be used for model prediction capabilities
(Brenning, 2005). The construction procedure of prediction rate is similar to the
success rate which the testing dataset (were not used in the training phase) has been
used for instead of training dataset. The area under the curve (AUC) of success and
prediction rate is the base for evaluation of model prediction power or assessment
accuracy of the groundwater potential models quantitatively (Khosravi et al., 2016a;
Khosravi et al., 2016b; Pham et al., 2017b). The AUC value varies from 0.5 to 1; the
higher the AUC, the better the prediction capability of models.

**3.6. Inferential statistics**

3.6.1-Freidman test
As the conditioning factors have been classified into different classes, non-parametric
test has been used in the current study. Non-parametric statistical procedures such as
Freidman test (Friedman, 1937) have been used regardless of statistical assumptions
(Derrac et al., 2011) and do not need the data to be normally distributed. The main aim
of this test is to find whether there is a significant difference between the performed
models or not. In other words, performing multiple comparisons to detect significant
differences between the behaviors of two or more models (Beasley and Zumbo, 2003).
The null hypothesis (H0) is that there are no differences among the performance of the
groundwater potential models. The higher the P-value, the higher the probability that
the null hypothesis is not true since if the p-value is less than the significance level
(α=0.05), the null hypothesis will be rejected.
3.6.2 Wilcoxon signed-rank test
The most important drawback of Freidman test is that it only illustrates whether there
is any difference between the models or not, and does not have the ability to show
pairwise comparisons among performed model. Therefore, another non-parametric
statistical test named Wilcoxon signed-rank test have been performed. To evaluate the
significance of differences between the performed groundwater potential models, the P
value and Z value have been used.





## 4. Result and analysis

### 4.1. Multi-collinearity diagnosis

Result of multi-collinearity analysis is shown in Table 1. Result has revealed that as VIF is less than 5 and the tolerance is greater than 0.1, there isn't any multi-collinearity problem among conditioning factors and all of factors are independent.

Table.1. Multi-collinearity analysis for conditioning factors

| No | Groundwater conditioning factors | Collinearity Statistics | |
| --- | --- | --- | --- |
| | | Tolerance | VIF |
| 1 | Slope degree | 0.231 | 2.401 |
| 2 | Slope aspect | 0.206 | 4.270 |
| 3 | Altitude | 0.801 | 2.097 |
| 4 | Curvature | 0.513 | 1.446 |
| 5 | SPI | 0.410 | 1.689 |
| 6 | TWI | 0.541 | 2.113 |
| 7 | TRI | 0.328 | 1.939 |
| 8 | Distance from fault | 0.408 | 2.25 |
| 9 | Distance from river | 0.212 | 3.126 |
| 11 | Landuse | 0.296 | 3.891 |
| 12 | Rainfall | 0.298 | 1.686 |
| 13 | Soil order | 0.205 | 4.039 |
| 10 | Geology (Unit) | 0.215 | 4.150 |

### 4.2. Spatial relationship between springs and the conditioning factors by SWARA method

The spatial correlation between springs and the conditioning factor has been shown in Table 2. For the slope, the class of 0-5.5 degree shows the highest probability (0.45) on spring groundwater occurrences and there is a contrary correlation between slope degree and SWARA values. As the slope degree increases, the probability of spring occurrence has reduced. In the case of slope aspect, the east aspect (0.44) has the most impact on spring occurrences followed by north (0.22), west (0.177), south (0.15) and flat (0.12) in the Koohdasht- Nourabad plain. According to calculated results, in terms of altitude, the springs are the most abundant in the altitude of 1703-2068 m (0.6) and the least abundant in the altitude of 1070-1385 m (0.04). The SWARA model is high in flat areas (0.4), followed by concave (0.38) and convex (0.2). For SPI, the highest SWARA value is found for the classes of 583969-1330153 (0.46), followed by the classes of 227099-583969(0.0.23) and 48664-227099 (0.19). In the case of the TWI, the SWARA values decrease when the TWI reduces, while the highest TWI belongs to the classes of 6.6-7.9 (0.47), and the lowest is for 2.1-4.6 (0.02). There is an adverse relationship between TRI and SWARA value, and

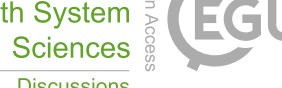

as the TRI increases, the SWARA value reduces. The highest and the lowest values of SWARA
also belongs to classes 0-8.7 (0.54) and 46.6-185 (0.001), respectively. For distance from the fault,
distance less than 2000 m has the highest impact on spring occurrences and with increase in the
distance (greater than 2000 m), the probability of spring occurrences has reduced. The highest
SWRA value belongs to distance from the fault of 500-1000 m (0.29) and the lowest value is for
greater than 2000 m (0.1). For the distance to river, it can be seen that the class of 0-200 m has the
highest correlation with the spring occurrence (0.46) and there is a contrary relationship between
spring occurrence and SWARA values; as the more the distance from the river, the lower the spring
occurrence probability. In the case of land use, the highest SWARA values are shown for garden
areas (0.219), followed by mixture of garden and agriculture (0.17), agricultural areas (0.12),
whereas the lowest SWARA is for bare soil and rock (0.00063). The rainfall between 500 and 600
mm has the highest SWARA value with 0.61 and the lowest SWARA belongs to 300-400 mm
(0.02). The Inceptisols have the highest SWARA values (0.5) followed by rock outcrop/Entisols
(0.39), rock outcrop/Inceptisols (0.056), Inceptisoils/Vertisoils (0.028), and Badlands (0.014). The
highest probability respectively belongs to the highly porous and very good water reservoir karstic
oligomiocene and cretaceous pure carbonate formation (OMq and K1bl), the young and poorly
consolidated highly porous detrital rock units (PeEf and Plq) and the unconsolidated quaternary
alluvium (PlQc).
Table.2. Spatial correlation between conditioning factors and the spring locations by SWARA methods

| Factors | Classes | Comparative importance of average value Kj | Coefficient Kj=Sj +1 | wj=(X(j-1))/kj | weight wj/ sigma wj |
|---|---|---|---|---|---|
| Slope (degree) | 0 - 5.55 | | 1.000 | 1.000 | 0.454 |
| | 5.55 - 12.11 | 0.300 | 1.300 | 0.769 | 0.349 |
| | 12.11 - 19.43 | 1.500 | 2.500 | 0.308 | 0.140 |
| | 19.43 - 28.77 | 2.000 | 3.000 | 0.103 | 0.047 |
| | 28.77 - 64.37 | 3.500 | 4.500 | 0.023 | 0.010 |
| Slope aspect | East | | 1.000 | 1.000 | 0.448 |
| | North | 1.000 | 2.000 | 0.500 | 0.224 |
| | West | 0.300 | 1.300 | 0.385 | 0.172 |
| | South | 0.100 | 1.100 | 0.350 | 0.156 |
| | Flat | 0.8 | 1.05 | 0.31 | 0.121 |
| Altitude (m) | 1703 - 2068 | | 1.000 | 1.000 | 0.608 |
| | 1385 - 1703 | 2.200 | 3.200 | 0.313 | 0.190 |
| | 2068 - 3175 | 0.800 | 1.800 | 0.174 | 0.106 |
| | 531 - 1070 | 1.000 | 2.000 | 0.087 | 0.053 |
| | 1070 - 1385 | 0.200 | 1.200 | 0.072 | 0.044 |





| | | | | | |
|---|---|---|---|---|---|
| Curvature | Flat | | 1.000 | 1.000 | 0.408 |
| | concave | 0.050 | 1.050 | 0.952 | 0.388 |
| | convex | 0.900 | 1.900 | 0.501 | 0.204 |
| SPI | 583969.72 - 1330153.27 | | 1.000 | 1.000 | 0.466 |
| | 227099.33 - 583969.72 | 1.000 | 2.000 | 0.500 | 0.233 |
| | 48664.14 - 227099.33 | 0.200 | 1.200 | 0.417 | 0.194 |
| | 0 - 48664.14 | 1.000 | 2.000 | 0.208 | 0.097 |
| | 1330153.27 - 4136452.25 | 10.000 | 11.000 | 0.019 | 0.009 |
| TWI | 6.64 - 7.92 | | 1.000 | 1.000 | 0.471 |
| | 5.60 - 6.64 | 0.700 | 1.700 | 0.588 | 0.277 |
| | 7.92 - 11.97 | 1.300 | 2.300 | 0.256 | 0.120 |
| | 4.63 - 5.60 | 0.100 | 1.100 | 0.233 | 0.110 |
| | 2.12 - 4.63 | 4.000 | 5.000 | 0.047 | 0.022 |
| TRI | 0 - 5.59 | | 1.000 | 1.000 | 0.544 |
| | 5.59 - 12.66 | 0.800 | 1.800 | 0.556 | 0.302 |
| | 12.66 - 20.62 | 1.500 | 2.500 | 0.222 | 0.121 |
| | 20.62 - 30.93 | 3.000 | 4.000 | 0.056 | 0.030 |
| | 30.93 - 75.13 | 10.000 | 11.000 | 0.005 | 0.003 |
| Distance from fault (m) | 0 - 200 | | 1.000 | 1.000 | 0.242 |
| | 200 - 500 | 0.050 | 1.050 | 0.952 | 0.231 |
| | 500 - 1000 | 0.100 | 1.100 | 0.866 | 0.210 |
| | 1000 - 2000 | 0.050 | 1.050 | 0.825 | 0.200 |
| | > 2000 | 0.700 | 1.700 | 0.485 | 0.118 |
| Distance from river (m) | 0 - 200 | | 1.000 | 1.000 | 0.464 |
| | 200 - 500 | 1.900 | 2.900 | 0.345 | 0.160 |
| | 500 - 1000 | 0.050 | 1.050 | 0.328 | 0.152 |
| | 1000 - 2000 | 0.300 | 1.300 | 0.253 | 0.117 |
| | > 2000 | 0.100 | 1.100 | 0.230 | 0.107 |
| Land-use | Garden | | 1.000 | 1.000 | 0.219 |
| | mixture of garden and agriculture | 0.282 | 1.282 | 0.780 | 0.171 |
| | agriculture | 0.340 | 1.340 | 0.582 | 0.128 |

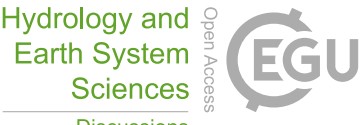

| | | | | | |
|---|---|---|---|---|---|
| | mixture of poor rangeland and follow | 0.419 | 1.419 | 0.410 | 0.090 |
| | follow | 0.233 | 1.233 | 0.333 | 0.073 |
| | mixture of moderate rangeland and agriculture | 0.294 | 1.294 | 0.257 | 0.056 |
| | mixture of very poor forest | 0.124 | 1.124 | 0.229 | 0.050 |
| | mixture of waterway and vegetation | 0.549 | 1.549 | 0.148 | 0.032 |
| | moderate forest | 0.205 | 1.205 | 0.122 | 0.027 |
| | mixture of agriculture with dry farming | 0.064 | 1.064 | 0.115 | 0.025 |
| | wood-land | 0.030 | 1.030 | 0.112 | 0.024 |
| | good rangeland | 0.043 | 1.043 | 0.107 | 0.023 |
| | rangeland | 0.333 | 1.333 | 0.080 | 0.018 |
| | poor rangeland | 0.030 | 1.030 | 0.078 | 0.017 |
| | poor forest | 0.210 | 1.210 | 0.065 | 0.014 |
| | moderate rangeland | 0.281 | 1.281 | 0.050 | 0.011 |
| | bare soil and rock | 0.237 | 1.237 | 0.041 | 0.009 |
| | dense rangeland | 0.278 | 1.278 | 0.032 | 0.007 |
| | dense-forest | 10.000 | 11.000 | 0.003 | 0.001 |
| | waterway | 0.000 | 1.000 | 0.003 | 0.001 |
| | mixture of agriculture with poor-garden | 0.000 | 1.000 | 0.003 | 0.001 |
| | very poor forest | 0.000 | 1.000 | 0.003 | 0.001 |
| | mixture of moderate forest and agriculture | 0.000 | 1.000 | 0.003 | 0.001 |
| | mixture of low forest and follow, | 0.000 | 1.000 | 0.003 | 0.001 |
| | urban and residential | 0.000 | 1.000 | 0.003 | 0.001 |
| | 600 - 700 | | 1.000 | 1.000 | 0.617 |
| | 700 - 800 | 2.200 | 3.200 | 0.313 | 0.193 |
| Rainfall (mm) | 800 - 900 | 0.600 | 1.600 | 0.195 | 0.121 |
| | 500 - 600 | 1.500 | 2.500 | 0.078 | 0.048 |
| | 400 - 500 | 1.300 | 2.300 | 0.034 | 0.021 |
| Soil order | Rock Outcrops/Entisols | | 1.000 | 1.000 | 0.509 |





| | | | | | |
|---|---|---|---|---|---|
| | Rock Outcrops/Inceptisols | 0.300 | 1.300 | 0.769 | 0.392 |
| | Inceptisols | 5.900 | 6.900 | 0.111 | 0.057 |
| | Inceptisols/Vertisols | 1.000 | 2.000 | 0.056 | 0.028 |
| | Bad Lands | 1.000 | 2.000 | 0.028 | 0.014 |
| | OMq | | 1.000 | 1.000 | 0.133 |
| | PeEf | 0.309 | 1.309 | 0.764 | 0.101 |
| | PlQc | 0.253 | 1.253 | 0.610 | 0.081 |
| | K1bl | 0.113 | 1.113 | 0.548 | 0.073 |
| | Plc | 0.014 | 1.014 | 0.541 | 0.072 |
| | pd | 0.059 | 1.059 | 0.511 | 0.068 |
| | TRKubl | 0.223 | 1.223 | 0.417 | 0.055 |
| | TRJvm | 0.027 | 1.027 | 0.406 | 0.054 |
| | MPlfgp | 0.048 | 1.048 | 0.388 | 0.051 |
| | OMql | 0.015 | 1.015 | 0.382 | 0.051 |
| | Plbk | 0.081 | 1.081 | 0.353 | 0.047 |
| | E2c | 0.291 | 1.291 | 0.274 | 0.036 |
| | TRKurl | 0.059 | 1.059 | 0.258 | 0.034 |
| Lithology (unit) | Qft2 | 0.335 | 1.335 | 0.194 | 0.026 |
| | MuPlaj | 0.100 | 1.100 | 0.176 | 0.023 |
| | KEpd-gu | 0.080 | 1.080 | 0.163 | 0.022 |
| | Kgu | 0.566 | 1.566 | 0.104 | 0.014 |
| | Qft1 | 0.064 | 1.064 | 0.098 | 0.013 |
| | Ekn | 0.109 | 1.109 | 0.088 | 0.012 |
| | KPeam | 0.027 | 1.027 | 0.086 | 0.011 |
| | PeEtz | 0.328 | 1.328 | 0.065 | 0.009 |
| | Kbgp | 0.445 | 1.445 | 0.045 | 0.006 |
| | EMas-sb | 0.310 | 1.310 | 0.034 | 0.005 |
| | Mgs | 0.626 | 1.626 | 0.021 | 0.003 |
| | TRJlr | 10.000 | 11.000 | 0.002 | 0.000 |
| | Klsol | 0.000 | 1.000 | 0.002 | 0.000 |
| | JKbl | 0.000 | 1.000 | 0.002 | 0.000 |

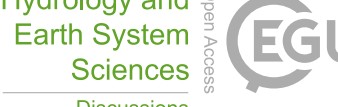



| | | | | |
|---|---|---|---|---|
| Kur | 0.000 | 1.000 | 0.002 | 0.000 |
| OMas | 0.000 | 1.000 | 0.002 | 0.000 |
| Mmn | 0.000 | 1.000 | 0.002 | 0.000 |


### 4.3. Application of ANFIS ensemble models and model's assessment

In the current study, hybrids of ANFIS model and five meta-heuristic algorithms were designed,
constructed and improved in MATLAB 8.0 software. These models are trained according to the
data of other intelligent models and the amount of training and optimization is tested by using
other data. All thirteen spring occurrence conditioning factors and the training dataset were applied
in building the model. Methods of these models are like this: gained weights by SWARA method
for each conditioning factor was fed as the input Training dataset was used for finding the
correlation between SWARA values of conditioning factor and springs (were assigned to 1), and
non-springs (were assigned to 0). These weights entered into a hybrid model as an output. It can
find and model the relationships between input and output data and the modeling accuracy is
calculated by statistical methods. The prediction ability of the five hybrid models with training
dataset as a target and estimated springs pixel as an output (in a training phase) and testing dataset
(in a validation phase) was shown in Fig.7 and Fig.8.
The MSE parameter indicates how much output of each hybrid's model is close to real rate. As it
can be seen in Fig. 4, MSE values of ANFIS-IWO, ANFIS-DE, ANFIS-FA, ANFIS-PSO, and
ANFIS-BA have been calculated for the training step 0.066, 0.066, 0.066, 0.049, and 0.09,
respectively. This shows that compared to other models, ANFIS-PSO had the best performance
while ANFIS-BA had the worst one for training step. However, it should be noted that training
step is not adequate for determining the best model for MSE optimization, and MSE level for
testing phase needs to be reviewed. According to the results shown in Fig.7, values of MSE –
0.060, 0.060, 0.060, 0.045, and 0.09 – relate to the hybrid models; ANFIS-IWO, ANFIS-FA,
ANFIS-PSO, and ANFIS-BEE have been calculated and indicate that the best performance is for
ANFIS-PSO, the worst for ANFIS-BA.





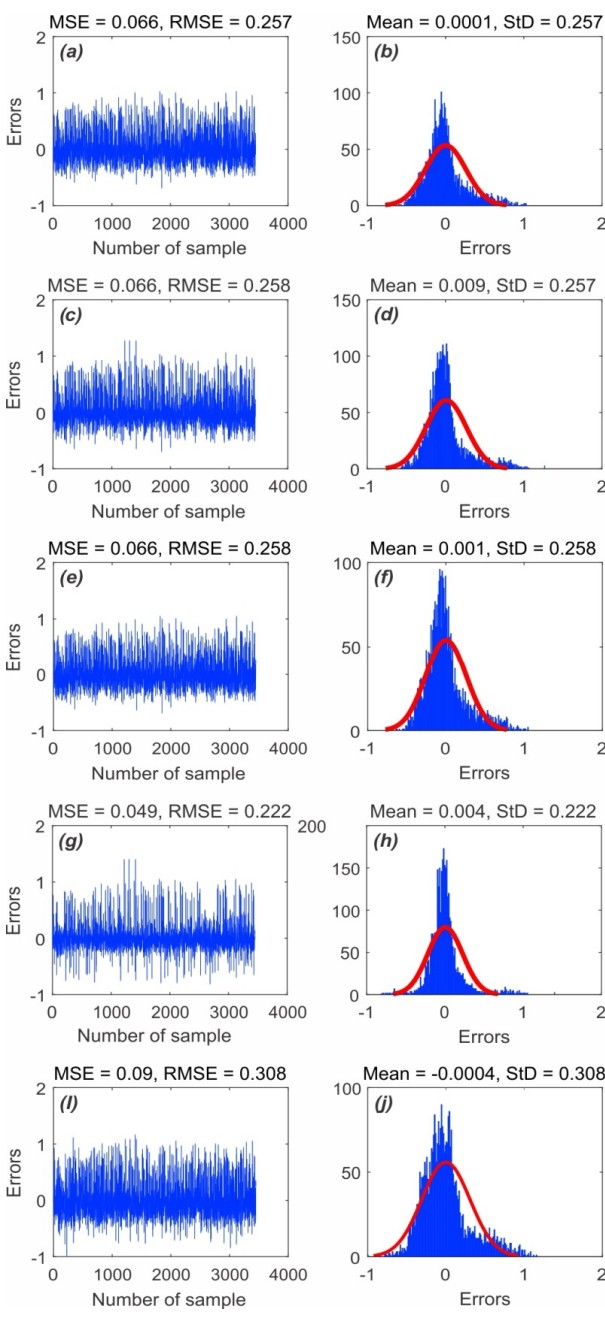


Fig. 7. MSE and RMSE values of the training data samples: a) ANFIS-IWO, c) ANFIS-DE, e) ANFIS-
FA, g) ANFIS-PSO l) ANFIS-BA frequency errors of train data samples of b) ANFIS-IWO, d) ANFIS-
DE, f) ANFIS-FA, h) ANFIS-PSO j) ANFIS-BA





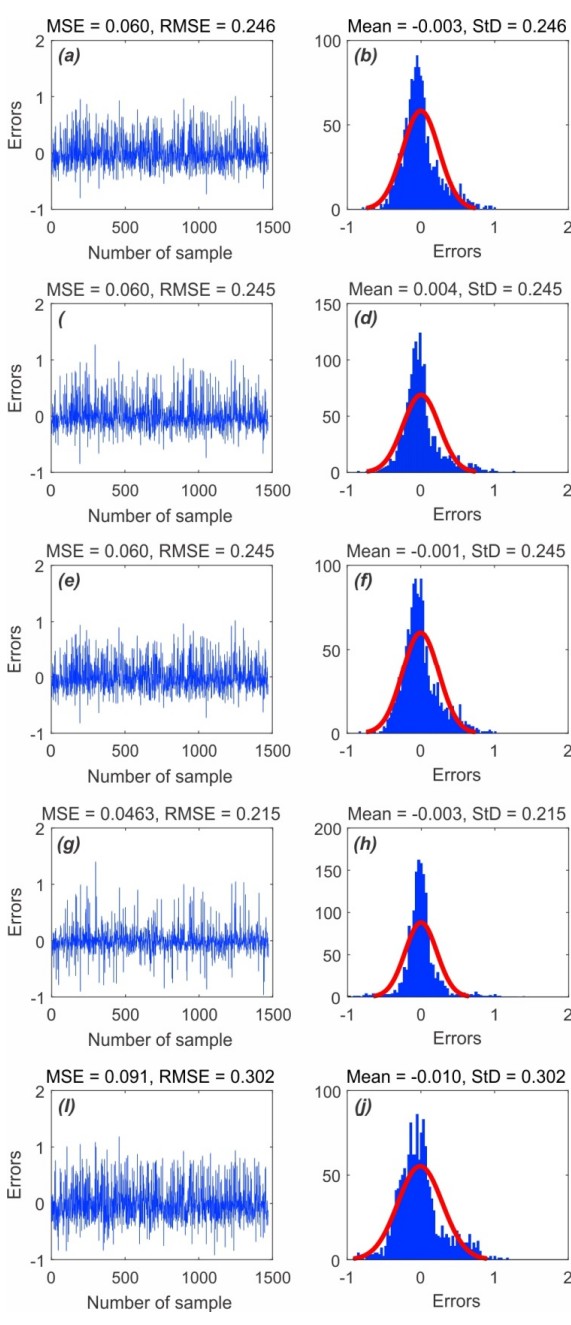


Fig.8. MSE and RMSE values of the validation data samples of a) ANFIS-IWO, c) ANFIS-DE, e)
ANFIS-FA, g) ANFIS-PSO l) ANFIS-BA frequency errors of test data samples of b) ANFIS-IWO, d)
ANFIS-DE, f) ANFIS-FA, h) ANFIS-PSO j) ANFIS-BA



However, it must be noticed that in addition to accuracy, determining the speed of used models
has recently found significance. To accomplish this, therefore, the processing time of 1000
iteration is calculated for each model where the amounts of 8036, 547, 22111, 1050, and 6993
seconds are related to ANFIS-IWO, ANFIS-DE, ANFIS-FA, ANFIS-PSO, and ANFIS-BA,
respectively (Fig. 9). As a result, it can be concluded that ANFIS-DE has had the minimum time
of processing speed compared to other models and ANFIS-FA has had the maximum time.

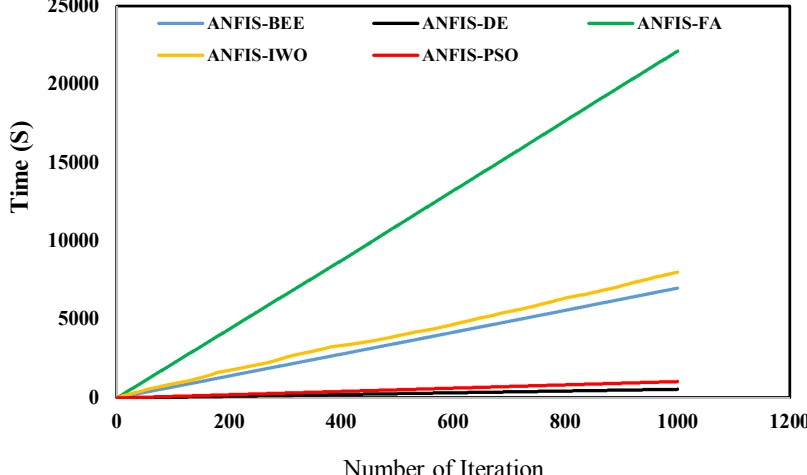


Fig. 9. Cumulative curve for speed processing of methods

On the other hand, it is possible to test how each model achieves convergence in learning. By
drawing a diagram, cost function values have been calculated in each iteration of convergence
graph for all five models as depicted in Fig.10. The results show that cost function values of
ANFIS-DE and ANFIS-BA become constant in 30 and 95 iterations. This indicates a rapid
convergence of every model. On the other side, ANFIS-PSO, ANFIS-IWO, and ANFIS-FA
achieved convergence in 650, 650, and 360 iterations, respectively that indicates the low speed of
these methods in reaching convergence.





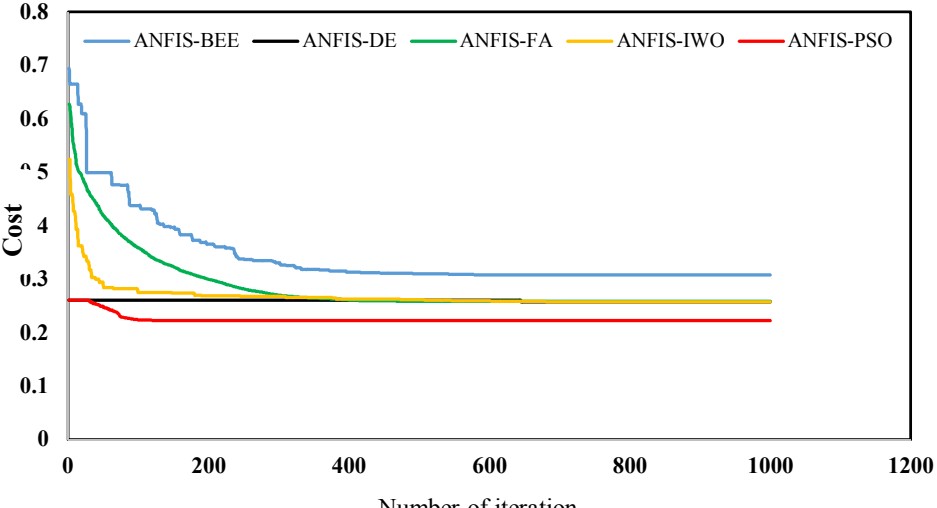


Fig.10. Convergence plot of methods

**4.4. Preparation of groundwater spring potential maps using ANFIS hybrid models**
In this study, SWARA values were standardized between 0-1 and were then transformed to
MATLAB software. Following that, ANFIS hybrid models of ANFIS with IWO, DE, FA, PSO
and BA algorithms were constructed using training dataset and standardized SWARA values. In
the next step, the built models were used for estimating the groundwater spring index (GSI), which
was assigned to whole the pixels of the study area and finally, the groundwater spring potential
mapping was developed from groundwater spring index. At first, each pixel was assigned to a
unique groundwater spring index. In second step, all indices were exported in ArcGIS10.2
software and were utilized in the construction of the groundwater spring potential mapping.
Ultimately, the archived maps were divided into five potential classes, namely very low, low,
moderate, high and very high based on quantile classification scheme. Therefore, based on the five
hybrid model, five maps of groundwater spring potential were prepared (Fig.11 a-e). There are six
methods, namely manual, equal interval, geometric interval, quantile, natural break and standard
deviation for classification based on the different purposes. The selection of the best method
depends on the characteristics of the data and the distribution of the groundwater spring indexes
in a histogram (Ayalew and Yamagishi, 2005). If the distribution of the indexes in the histogram
is normal or close to normal, two methods of Equal interval and standard deviation are used.
However, if the indexes have a positive or negative skewness, the quantile or natural break
classification is proper for indexes classification (Akgun, 2012). In this research, the histogram
was checked and the results revealed that quantile method was better than other methods for
indexes classification.








Fig.11. Groundwater spring potential mapping using ANFIS-IWO (a), ANFIS-DE (b), ANFIS-FA (c), ANFIS-PSO
(d) and ANFIS-BA (e).






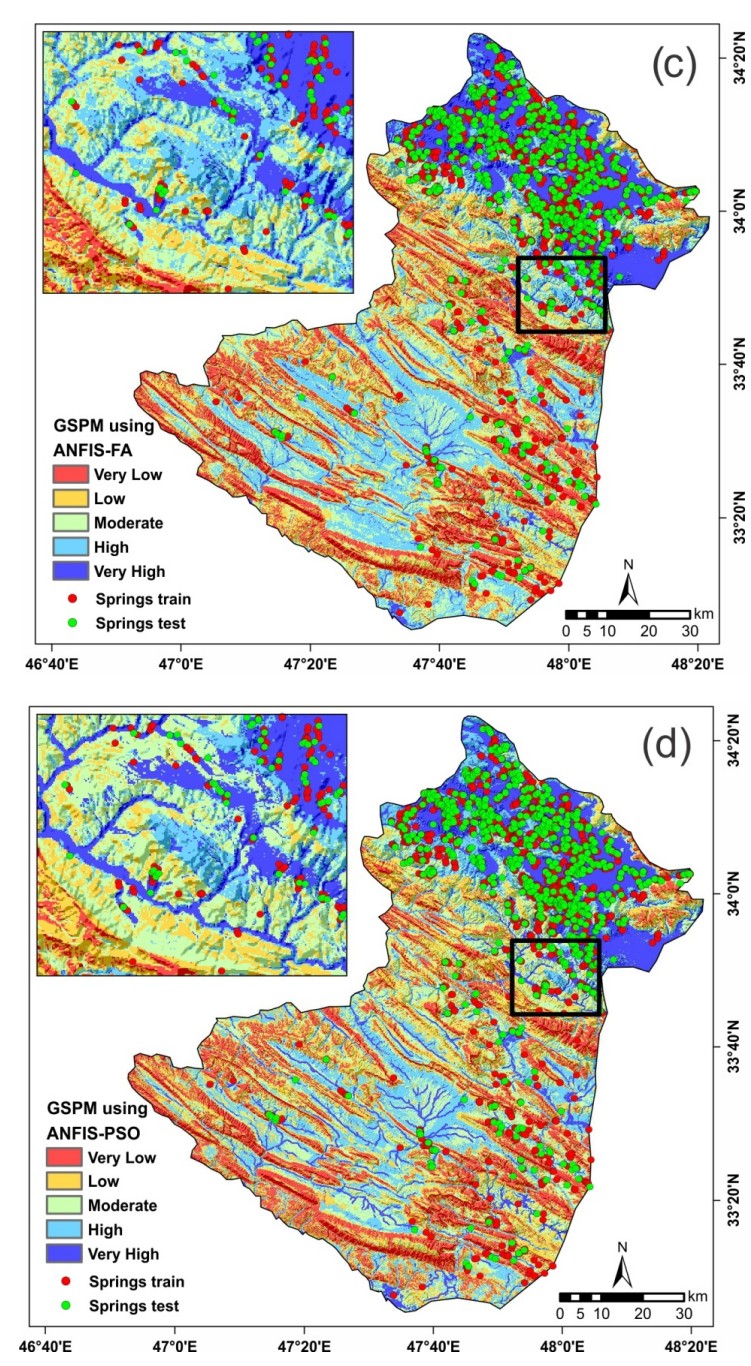


Fig.11. Continued





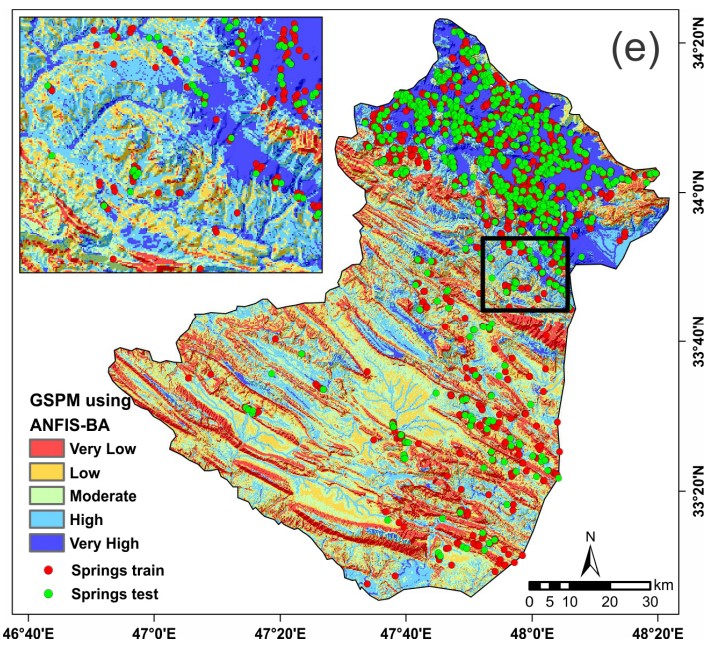


Fig.11. Continued


## 4.5. Validation and comparisons of the groundwater spring potential map

The prediction ability and reliability of the five achieved maps have been evaluated by both
training and testing dataset. The results of the success rate revealed that the ANFIS-DE had the
highest AUC value of 0.883 followed by ANFIS-IWO and ANFIS-FA (0.882), ANFIS-PSO
(0.871) and ANFIS-BA (0.852) (Fig.12a). The results exhibited that all five models had a very
good prediction capability but the ANFIS-DE has the highest prediction rate (0.873) followed by
NFIS-IWO and ANFIS-FA (0.873), ANFIS-PSO (0.865) and ANFIS-BA (0.839), respectively
(Fig.12b).

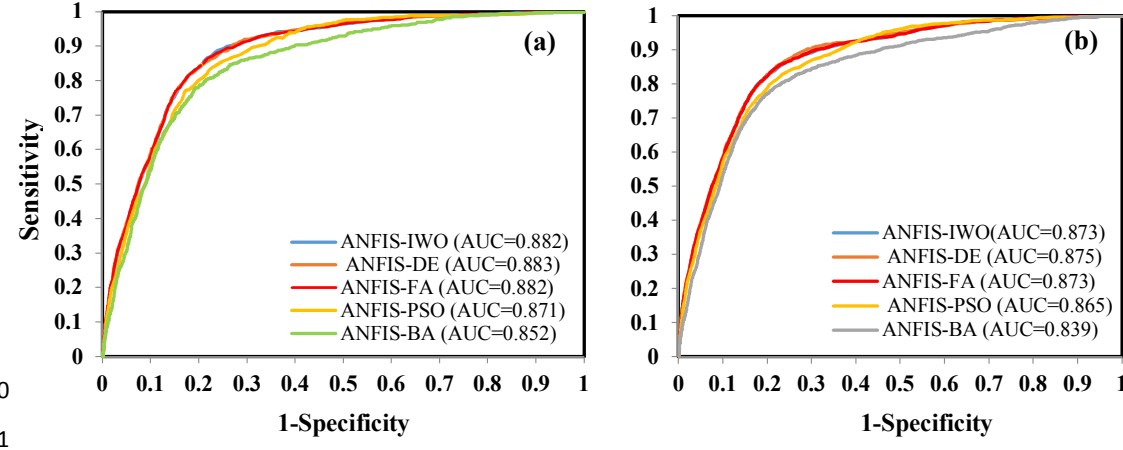





Fig.12. Success rate (a) and prediction rate (b) curves for the five performed models

### 4.6. Non-parametric statistical tests

The two tests of Freidman and Wilcoxon signed rank have been performed to determine whether
there are any statistically significant differences between the models performance or not. The result
of Freidman test revealed that (Table.3) as Sig and chi-square values were less than 0.05 and
greater than 3.84, respectively, null hypothesis has been rejected. The result also indicated that
there was statistically a significant difference between prediction capabilities of these five models.

Table.3. The result of Freidman test

| NO | Performed models | Mean rank | Chi-square | Sig |
|----|------------------|-----------|------------|-----|
| 1 | ANFIS-DE | 3.04 | | |
| 2 | ANFIS-IWO | 3.13 | | |
| 3 | ANFIS-FA | 2.98 | 64.84 | 0.00 |
| 4 | ANFIS-PSO | 2.72 | | |
| 5 | ANFIS-BA | 3.12 | | |


To show the pairwise differences between models performance, the Wilcoxon signed rank test was
carried out and result were shown in Table 4. Result of the Wilcoxon signed-rank test showed that
both P-values and $z$ were far from the standard values of 0.05 and (from -1.96 to + 1.96),
respectively except for ANFIS-FA vs. ANFIS-DE and ANFIS-PSO vs. ANFIS-DE. This indicates
that there are statistically significant differences between models performance except for ANFIS-
FA vs. ANFIS-DE and ANFIS-PSO vs. ANFIS-DE.

Table.4. The result of Wilcoxon signed rank test

| NO | Pairwise comparison | Z-Value | P-Value | Significance |
|----|---------------------|---------|---------|--------------|
| 1 | ANFIS-DE vs. ANFIS-BA | -3.97 | 0.00 | Yes |
| 2 | ANFIS-FA vs. ANFIS-BA | -2.37 | 0.017 | Yes |
| 3 | ANFIS-IWO vs. ANFIS-BA | -2.35 | 0.018 | Yes |
| 4 | ANFIS-PSO vs. ANFIS-BA | -3.04 | 0.002 | Yes |
| 5 | ANFIS-FA vs. ANFIS-DE | -1.32 | 0.185 | No |
| 6 | ANFIS-IWO vs. ANFIS-DE | -3.96 | 0.00 | Yes |
| 7 | ANFIS-PSO vs. ANFIS-DE | -0.841 | 0.41 | NO |
| 8 | ANFIS-IWO vs. ANFIS-FA | -3.19 | 0.001 | Yes |
| 9 | ANFIS-PSO vs. ANFIS-FA | -1.90 | 0.057 | Yes |



| 10 | ANFIS-PSO vs. ANFIS-IWO | -2.44 | 0.015 | Yes |
|----|--------------------------|-------|-------|-----|


### 4.7. Percentage area

The percentage area of each class of final map resulting from five hybrid models has been represented in Fig.13. According to results, as ANFIS-DE is more accurate in groundwater spring prediction capabilities, the percentage areas of very low, low, moderate, high and very high groundwater spring potential are about 19.06, 19.88, 21.72, 20.55 and 18.78 % of the study area, respectively.

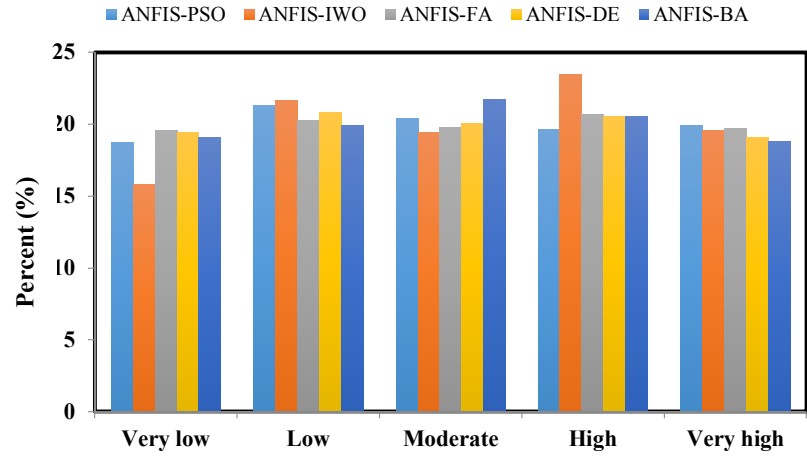

.

Fig.13. Percentage areas of different groundwater spring potential classes for five models

### 5. Discussion

As the most important natural resources in the world, groundwater is estimated to form around 25 percent of the all fresh water (Alley et al., 1999). Groundwater potential assessment has been considered as one of the effective methods in the management of groundwater resources regarding its better exploitation and conservation (Naghibi et al., 2017) and the prediction accuracy of the achieved maps depend on the method used. Koohdasht-Nourabad plain in Lorestan province is one of the most important plains in the country since majority of people in the area are farmers and their water needs are met through groundwater extraction. Therefore, in the present study five novel hybrid models have been applied for (1) identifying the groundwater spring potential mapping with a high precision and, (2) comparing prediction capabilities of these five different hybrid models in groundwater potential modeling. For achieving the aim of the current research, five hybrid models of ANFIS-Invasive Weed Optimization, ANFIS-Differential Evolution, ANFIS-Firefly, ANFIS-Particle Swarm Optimization and ANFIS-Bees algorithms had been used.

### 5.1. The impact of conditioning factor's classes on GSPM

The classification of conditioning factor is a necessary step in finding the correlation analysis between spring and conditioning factor. It should be noted that there isn't any universal guideline



for the number and size of the classes as well as selecting the conditioning factors and they mostly
depend on some factors including characteristics of the study area and previous similar studies (Xu
et al., 2013). As the slope increase, the probability of the water infiltration reduces and runoff
generation will increase. Thus, the more the slope, the lowest the spring occurrence probability.
According to the result of the SWARA method, the springs almost occur in a middle altitude or
mountain slopes (but wells are dug in a low-land area). The flat curvature retains and infiltrates
rainfall. Therefore, the amount of groundwater in these areas is higher than concave or convex
curvature. The east aspect has more springs than other aspects. These results are in accordance
with Pourtaghi and Pourghasemi (Pourtaghi and Pourghasemi, 2014), that had explained most
springs occurred in the elevation of 1600-1900 m and east slope aspect (with FR method). TWI
shows the amount of wetness, and it is obvious that the more the TWI, the higher the springs
probability occurrence is. Terrain Roughness Index (TRI) or topographic roughness or terrain
ruggedness calculates the sum of change in elevation between a grid cell and its neighborhood,
and as the lowest the roughness, the highest spring potential mapping. The SPI shows the erosive
power of the water and mountainous area is higher than plain area. So, As the SPI increases, the
spring potential occurrence increases. Rivers are one of the most important sources of groundwater
recharge and the nearer to river, the higher probability to springs occurrences. Also, as the rainfall
increases, the higher springs incident, but in the current study, some other conditioning factors
affected the spring occurrences.
Most of the springs were located in the garden land-use. Therefore, it can be stated that the gardens
have been established near the springs. Pliocene-Quaternary formation in a geologic time scale is
newer and Quaternary formation has a high potential to groundwater springs incident due to high
permeability. The fault is discontinuity in a volume of rock. Thus, the nearer to the fault, the higher
the spring occurrence probability will be. Inceptisols soils are relatively new and are characterized
by having only the weakest appearance of horizons, the most abundant on the Earth
(https://www.britannica.com/science/Inceptisol) and mostly formed from colluvial and alluvial
materials. So, due to high permeability and high rainfall infiltration, they have a high potential for
springs occurrences. In the case of lithological unit, there are four suitable rock type as water
reservoir based on physical phenomena such as porosity and permeability that consist of: 1.
unconsolidated sands and gravels; 2. sandstones; 3. Lime-stones; and 4. basaltic lava flows. In this
study area lithological units include sedimentary rocks mostly carbonate and detrital rocks with
cover of alluvium and minor soil.
**5.2. Advantages/disadvantages of the models and performance analysis**
The highest accuracy based on the RMSE in both training and testing dataset belonged to ANFIS-
PSO, but based on the AUC for success and prediction rate, the ANFIS-DE had the highest
prediction capability. The problem with RMSE comes from the fact that, it is based on the error
assessment. But the models should be acted upon holistically based on the abilities. AAUC for
Receiver operating characteristic (ROC) curves (success and prediction rate curves) is based on
the true positive (TP), true negative (TN), false positive (FP) and false negative (FN), it is more
accurate than RMSE for comparison (Termeh et al., 2018). The two axes of the ROC curves are
(Negnevitsky, 2005):
$X = 1 - \text{specificity} = 1 - (TN/(TN + FP))$          (24)





$Y = \text{sensitivity} = (TP/(TP/FN))$ (25)
ANFIS model is one of the machine learning algorithms that is proper for natural phenomenon
modeling due to its non-linear structure. The ANFIS model, which is based on Takagi–Sugeno
fuzzy inference system, is a hybrid of ANNs and fuzzy logic. Therefore, it has a potential to
capture the benefits of both in a single framework and can be considered as a robust model. The
predictions in ANFIS model are based on learning the ''if–then'' rules between groundwater
spring locations and conditioning factors.
Polykretis et al. (Polykretis et al., 2017), applied ANFIS for landslide susceptibility mapping
(LSM) in Peloponnese peninsula, Grece and stated that ANFIS model was a robust model.
Vahidinia et al. (Vahidnia et al., 2010), applied ANFIS model to LSM in the Mazandaran Province,
Iran, and revealed that ANFIS was a flexible and non-linear model and was completely appropriate
for building a framework of easy inferences. Isanta Navarro (Isanta Navarro, 2013), applied
ANFIS to stability augmentation of an airplane and stated that ANFIS had some advantages
including: (1) much better learning ability, (2) need for fewer adjustable parameters than those
required in other neural network structure and (3) allowing a better integration with other control
design methods by its networks.
Despite several advantages of ANFIS model, non-adjutancy of membership function is the biggest
disadvantage of this model. Finding the optimal parameter for neural fuzzy model in a membership
function is difficult; therefore, the best parameter should be finding other optimization models.
This problem was addressed in this paper for being solved by five meta-heuristic algorithms,
namely Invasive Weed Optimization, Differential Evolution, Firefly, Particle Swarm Optimization
and Bees algorithms. The aim of any optimization is to find values of the variable to gratify the
restriction by minimizing or maximizing the objective function. These optimization algorithms are
completely new in environmental modeling (especially in groundwater potential mapping) and
have been used for natural hazards assessment by a few researchers in landslide susceptibility
assessment (Chen et al., 2017a) as well as in flood susceptibility mapping (Bui et al., 2016; Termeh
et al., 2018).
In the current study, the results showed that DE algorithm optimized the parameter for neural fuzzy
model better than four other algorithms. The main DE algorithm's advantage is its simplicity as it
consists of only three parameters called N (size of population), F (mutation parameter) and C
(crossover parameter) for controlling the search process (Tvrdík, 2006). Advantages of DE
algorithm can be explained as follows: (1) Ability to handle non-differentiable, nonlinear and
multimodal cost functions, (2) Parallelizability to cope with computation intensive cost functions,
(4) good convergence properties, i.e. consistent convergence to the global minimum in consecutive
independent trials, and (5) random sampling and combining vectors in the present population for
creating vectors for the next generation.
Finally, it should be noted that each algorithm has some advantages or disadvantages according to
the optimization problems which can be summarized as:
Some of the advantages of IWO in comparison to other evolutionary algorithms include the way
of reproduction, spatial dispersal, and competitive exclusion (Mehrabian and Lucas, 2006) as well
as the fact that seeds and their parents are ranked together and those with better fitness survive and
become reproductive (Ahmed et al., 2014). This algorithm can benefit from combined advantages



of retaining the dominant poles and the error minimization (Abu-Al-Nadi et al., 2013) and there is
no need for continuity or differentiability of the objective function.
Bees algorithm doesn't employ any probability approach, but utilizes fitness evaluation to drive
the search (Yuce et al., 2013). This algorithm is implemented with several optimization problems
or in other words, BA uses a set of parameters including the number of scout bees in the selected
patches, the number of best patches in the selected patches, the number of elite patches in the
selected best patches, the number of recruited bees in the elite patches, the number of recruited
bees in the non-elite best patches, the size of neighborhood for each patch, the number of iterations
and the difference between the value of first and last iterations that makes it powerful. BA also has
both local and global search capability and the local search step of the algorithm covers the best
locations. BA is really easy to use and available for hybridization combination with other
algorithms (Yuce et al., 2013). Another advantage is hiring smart bees since bees (artificial insects)
can memorize the location of the best food source and its quality which has been found before. If
the new solution has a lower fitness than the best-saved solution in the SB memory, it is replaced
with new candidate solution (Gorji-Bandpy and Mozaffari, 2012).
Firefly Algorithm's (FA) advantages are summarized as: (1) handling highly non-linear, multi-
modal optimization problems efficiently, (2) not utilizing velocities (3) very high speed of
convergence in finding the global optimized answer (4) ability to be integrated with other
optimization techniques as a flexible method, and finally (5) not needing a good initial solution to
beginning of its iteration process.
Advantages of Particle Swarm Optimization (PSO) algorithm can be summarized as follows: (1)
Particles update themselves with the internal velocity; (2) particles have a memory important to
the algorithm, (3) the 'best' particle gives out the information to others, (4) it often produces quality
solutions more rapidly than alternative methods, (5) this algorithm simulates bird flocking
behavior to achieve a self-evolution system, (6) it automatically searches for the optimum solution
in the solution space, (7) (Wan, 2013).
As a result, there isn't any algorithm which works perfectly for all optimization problems, and
each algorithm has a different performance accuracy based on different data. New algorithms,
therefore, should be applied, tested and finally the most powerful algorithm should be selected; as
the conclusion of the research demands.
**5.3. Previous works and future work proposal**
Some research has been done in groundwater well or spring potential mapping using bivariate
statistical models (Al-Manmi and Rauf, 2016; Guru et al., 2017; Nampak et al., 2014) using
random forest (Rahmati et al., 2016) and using boosted regression tree and classification and
regression tree (Naghibi et al., 2016). The ANFIS-metaheuristic hybrid models are not used in
groundwater potential mapping and are only used in flood susceptibility mapping (Bui et al., 2016;
Termeh et al., 2018) and landslide susceptibility mapping (Chen et al., 2017a). Tien Bui et al. (Bui
et al., 2016) ensemble the ANFIS using two optimization models, namely Genetic (GA) and PSO
for the identification of flood prone areas in Vietnam. Razavi Termeh et al. (Termeh et al., 2018),
used ANFIS-Ant Colony Optimization, ANFIS-GA and ANFIS-PSO in flood susceptibility
mapping of Jahrom basin and stated that ANFIS-PSO had higher prediction capabilities than the
two other models. Chen et al (2017) applied three hybrid models, namely ANFIS- Genetic



Algorithm (GA), ANFIS-Differential Evolution (DE) and ANFIS-Particle Swarm Optimization
(PSO) for identifying the areas prone to landslides in Hanyuan County, China. The results showed
that ANFIS-DE had a higher performance (AUC=0.84) followed by ANFIS-GA (AUC=0.82) and
ANFIS-PSO (AUC=0.78).
Generally, the mentioned results of the present study and different researchers revealed that by
applying hybrid models, better results could be achieved for any spatial prediction modeling
including groundwater potential mapping. The ensembles of ANFIS by meta-heuristic algorithms
can be proposed for any spatial prediction modeling such as groundwater potential mapping, flood
susceptibility mapping, landslide susceptibility assessment, gully occurrences susceptibility
mapping and other endeavors at a regional scale and in other areas.
For future work, it is recommended that (1) the water quality of the Koohdasht-Nourabad plain be
investigated and the water quality of areas with high potential be determined for different aspects
such as drinking, agricultural and industrial activities, and (2) the groundwater vulnerability
assessment should be applied by some common methods including DRASTIC model for which
the zones with high potential to groundwater occurrences should be preserved against pollution.
**6. Conclusion**
Groundwater is the most important natural resource in the world and about 25 percent of all fresh
water is estimated as groundwater. Thus, the groundwater potential mapping has been considered
as one of the most effective methods for the management of groundwater resources for better
exploitation. The conservation and the maps with high accuracy is necessary for decisions. As the
natural phenomena are complex, the simple method and statistical models do not have an
appropriate result in modeling of the natural phenomena. To solve the problem, the artificial
intelligence models have been used for having a reasonable result but these model have some
weaknesses, especially in modeling process. To resolve this problem, this study verifies the five
new hybrid models of ANFIS with metaheuristic algorithms namely IWO, DE, FA, PSO and BA
to increase the prediction capability of the spatial prediction of groundwater potential mapping (1)
for solving the weakness of the artificial intelligence models and (2) using non-linear structure of
these models which are better for modeling of the complex natural phenomena such as
groundwater modeling. The result of this modeling has been evaluated using prediction rate ROC
curves and the results showed that all models had very good reasonable results. However, the
ANFIS-DE had the highest prediction power (0.875) followed by ANFIS-IWO and ANFIS-FA
(0.873), ANFIS-PSO (0.865) and ANFIS-BA (0.839). Thus, the results revealed that the
metaheuristic algorithms could optimize the weights parameters of the ANFIS model with high
accuracy as the highest advantage of these algorithms
According to the results of the SWARA method, most springs existed in an altitude of 1703-2068
m, flat curvature, east aspect, TWI of 6.6-7.9, TRI of 0-8.7, SPI of 583969-1330153, Inceptisols
soil, slope of 0-5.5 degree, 0-200 m distance from river, 500-1000 m distance from fault, rainfall
between 500-600 mm, in a garden, in a Pliocene-Quaternary lithological age and OMq lithology
unit.
The results of the current study is helpful for Iran Water Resources Management Company
(IWRMC) for sustainable management of the groundwater resources. Overall, the maps resulting
from these hybrid artificial intelligence algorithms can be applied for better management of the



groundwater resources in the study area, and can be used for other areas for groundwater potential
assessment or mapping of gully, flood, landslide and other susceptibility uses in the world due to
its high precision.

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
