# Peer review of "Spatial Prediction of Groundwater Spring Potential Mapping Based on"

_Hydrology and Earth System Sciences, 2017_

## Referee Comment (RC1) · Anonymous Referee #1 · 20 Feb 2018

Dear Editor This is great work; meanwhile I have the following moderate comments on above MS. 1. I think title of the mentioned research is very long; please authors try to decrease it. 2. In abstract, your means from curvature is which one? Plan or

profile? 3. In abstract, what is your means from soil order? 4. Results of models are very similar together. Please edit results of lines 33-35. 5. Please add a reference in lines 118-119 for rainfall descriptions. 6. Quality of Fig. 2 isn't proper. Please draw it again. 7. Please add source of groundwater spring inventory map. 8. Please explain about classification of different layers or at least add some citations for the mentioned classifications 9. Fig. 3 (j) and 3 (m) what are codes? 10. According to Table 2, I think it isn't a land use map, it is land cover. Please change its name. or present land use/ land cover

---

## Referee Comment (RC2) · Anonymous Referee #2 · 13 Mar 2018

This is an interesting manuscript to understand the spatial prediction of groundwater spring potential mapping using new hybrids of ANFIS. Some comments are provided as following.

1. There are many grammatical mistakes appeared in the article. I strongly request that this manuscript should be totally edited by professional English editors for improving the English writing. 2. In Figure 2, thirteen groundwater conditioning factors were served as input of hybrid models. The sensitivity analysis should be performed to investigate that which conditioning factor is most important factor to affect the output. 3. How many non-spring stations in the study area? Why are the numbers of non-spring stations should same with spring stations? 4. In line 171, "In the current study, 14 conditioning factors.." should be "13 conditioning factors..". 5. I am wondering that Figures 4, 5, and 6 are necessary, because these figures are taken from other literature. 6. In equation (1), what is "I"? The term "I" should be "i=1". 7. Pages 12, 13, and 14, the authors spend many spaces to describe the algorithm of ANFIS model. Actually, we can find the same description in many references. I am also wondering that the description of ANFIS model is necessary. Should it move to Appendix. 8. In "Discussion" Section, lines 708-719, why are these sentences put here? These sentences seem to repeat again. 9. Can the authors describe that how much time you spend to run for each hybrid ANFIS model in MATLAB environment? 10. The important factors to be adjusted for each hybrid ANFIS model should be listed with a Table,

---

## Author Comment (AC1) · 6 Apr 2018

Thank you so much for your positive and valuable comments. This document explains the changes made in the revised manuscript while dealing with the comments raised by the reviewers.

Comment 1: I think title of the mentioned research is very long; please authors try to decrease it.

Answer: Thanks for you valuable comment. Authors agree with you, the title has been shortened as follows: A comprehensive study of new hybrid models for ANFIS with five meta-heuristic algorithms (IWO, DE, FA, PSO, BA) for spatial prediction of groundwater spring potential mapping.

Comment 2: In abstract, your means from curvature is which one? Plan or profile?

Answer: Our mean is plan curvature which was corrected from throughout the paper.

Comment 3: In abstract, what is your means from soil order?

Answer: To identify, understand, and manage soils, soil scientists have developed a soil classification or taxonomy system. Like the classification systems for plants and animals, the soil classification system contains several levels of detail, from the most general to the most specific. The most general level of classification in the United States system is the soil order, of which there are 12 (such as Alfisols, Aridisoils, and etc.). Each order is based on one or two dominant physical, chemical, or biological properties that differentiate it clearly from the other orders.

Comment 4: Results of models are very similar together. Please edit results of lines 33-35.

Answer: Thanks for this valuable comment. The sentences have been corrected as : Although the results of performed models are close to each other, but ANFIS-DE has the highest prediction capability (0.875) for groundwater spring potential mapping in the study area, followed by ANFIS-IWO and ANFIS-FA (0.873), ANFIS-PSO (0.865) and ANFIS-BA (0.839).

Comment 5: Please add a reference in lines 118-119 for rainfall descriptions.

Answer: The proper references have been added as ''Lorestan Weather Bureau report, 2016"

Comment 6: Quality of Fig. 2 isn't proper. Please draw it again.

Answer: This Figure was draw again and added to the paper.

Comment 7: Please add source of groundwater spring inventory map

Answer: The proper source have been added: a total of 2463 springs were selected from documentary source (Iranian Water Resources Management) and considered for modeling.

Comment 8: Please explain about classification of different layers or at least add some citations for the mentioned classifications.

Answer: Some references have been added to the sentences as the following: The process of converting continuous variables into categorical classes were carried out using frequency analysis of springs location (Khosravi et al, 2018; Ahmadisharaf et al., 2016) in order to define the class intervals (Bui et al., 2011).

Comment 9: Fig. 3 (j) and 3 (m) what are codes?

Answer: Thank you for your precis attention. it was corrected and considered at the paper the authors corrected it on the paper in a simple way and avoid from the description on the table.

Comment 10: According to Table 2, I think it isn't a land use map, it is land cover. Please change its name or present land use/land cover

Answer: Thank you for your precis attention; it was corrected to land-use/land-cover throughout the paper.

Dear Editor and reviewers: Thank you so much for your viewpoints and comments in regarding our manuscript. I hope the emendations caused to consent the respected reviewer and editor-in-chief and made my paper well qualified for publication.

Please also note the supplement to this comment:
https://www.hydrol-earth-syst-sci-discuss.net/hess-2017-707/hess-2017-707-AC1-supplement.pdf

**Supplement:**

**A comprehensive study of new hybrid models for ANFIS with five meta-heuristic algorithms (IWO, DE, FA, PSO, BA) for spatial prediction of groundwater spring potential mapping**

Khabat Khosravi[1], Mahdi Panahi*[2], Dieu Tien Bui*[3]

1-Department of watershed management engineering, Faculty of Natural Resources, Sari Agricultural Science and Natural Resources University, Sari, Iran. (E-mail: khabat.khosravi@gmail.com)
2- Department of Geophysics, Young Researchers and Elites Club, North Tehran Branch, Islamic Azad University, Tehran, Iran. (E-mail: panahi2012@yahoo.com)
3- Geographic Information System Group, Department of Business and IT, University College of Southeast Norway, Gullbringvengen 36, 3800 Bø i Telemark, Norway. (E-mail: Dieu.T.Bui@usn.no)

**Abstract**
Groundwater is one of the most valuable natural resources in the world; therefore developing advanced tools for sustainable management of the groundwater is highly necessary. One of the most important tools for the management of the groundwater is groundwater potential map (GPM). The current study's aim is to proposed and verified new artificial intelligence methods for spatial prediction of groundwater spring potential mapping at Koohdasht-Nourabad plain, Lorestan province, Iran. These methods are new hybrids of Adaptive Neuro-Fuzzy Inference System (ANFIS) with five meta-heuristic algorithms, Invasive Weed Optimization (IWO), Differential Evolution (DE), Firefly (FA), Particle Swarm Optimization (PSO), and Bees (BA) algorithm. Accordingly, a total of 2463 springs were identified and collected, and then, divided in two subsets randomly, including 70% (1725 locations) of the total springs were used for training models, whereas the remaining 30% (738 spring locations) were utilized for the model evaluation. Thirteen groundwater conditioning factors, slope degree, slope aspect, altitude, plan curvature, stream power index (SPI), topographic wetness index (TWI), terrain roughness index (TRI), distance from fault, distance from river, land-use/land-cover, rainfall, soil order, and lithology were prepared for modeling. In the next step, the Stepwise Assessment Ratio Analysis (SWARA) method was employed to quantify the degree of relevance of these conditioning factors and the springs. The global performance of these derived models was assessed using the Area Under the curve (AUC). In addition, the Freidman and Wilcoxon signed rank test were carried out to check and confirm the best model in this study. The result showed that these models has high performance; however, the ANFIS-DE mdel has the highest prediction capability ( AUC = 0.875), followed by the ANFIS-IWO model, the ANFIS-FA model (0.873), the ANFIS-PSO model (0.865), and the ANFIS-BA model (0.839). The results of this research can be useful for decision makers to sustainable management of groundwater resources.

Key words: Groundwater spring, ANFIS-DE, ANFIS-IWO, ANFIS-FA, ANFIS-PSO, ANFIS-BA, Iran.

**1. Introduction**

Groundwater is defined as the water in a saturated zone which fills rock and pore spaces (Berhanu et al., 2014; Fitts, 2002), whereas groundwater potential is the possibility of groundwater occurrence in an area (Jha et al., 2010). The occurrence of groundwater in an aquifer is affected by various geo-environmental factors including lithology, topography, geology, fault and fracture
and its connectivity, drainage pattern and land-use/land-cover (Mukherjee, 1996). Geological
strata acts like a conduit and reservoir for groundwater whilestorage and transmissivity influence
the suitability of exploitation of groundwater in a given geological formation. Downhill and
depression slopes impart runoff and improve recharge and infiltration, respectively (Waikar and
Nilawar, 2014).

Groundwater, which serves as a major source of drinking water to communities, agricultural and
industrial sectors, is one of the most precious natural resources in the world (David Keith Todd
and Mays, 1980) due to its consistent temperature and widespread availability, low vulnerability
to pollution, low development cost, and drought dependability (Jha et al., 2007). Globally, 1.5
billion people are dependent on groundwater, solely for drinking purposes, and about 38% of the
irrigated lands depend on the groundwater itself (Siebert et al., 2013). Due to population growth, ,
the demand of water is constantly increased. A major challenge now is how to have sustainable
management system of groundwater to preserve and ensure continuous supply with regards to the
water demand. One of the most important measures for the groundwater resource management is
to collect adequate knowledge on spatial and temporal distribution of groundwater, its quantity as
well as its quality.

For the case of Iran, Approximately two-third of the land is covered by deserts. As a result, similar
to other arid regions, the main sources of water supply for drinking and other are the groundwater
(Nosrati and Van Den Eeckhaut, 2012). Agriculture, which is one of the most prominent economic
sectors in Iran, and especially, in the study area, is still be limited due to water scarcity (Zehtabian
et al., 2010). Groundwater in Iran supplies around 65% of the water use-up and the remaining 35%
is supplied by surface water (Rahmati et al., 2016). One of the most important measures to
responsible for the increase of fresh-water is to identify groundwater potential zoning, an essential
tool for performing a successful groundwater determination, protection, and management program
(Ozdemir, 2011a).

There are a number of methods for groundwater exploitation in traditional approaches including
drilling as well as geological, geophysical, and hydrogeological methods. Yet, they are time-
consuming, costly (David Keith Todd and Mays, 1980; Israil et al., 2006; Jha et al., 2010; Sander
et al., 1996; Singh and Prakash, 2002). Recently, the application of geographic information
systems (GIS) and remote sensing (RS) has become an effective procedure for groundwater
potential mapping (Fashae et al., 2014) due to their ability in handling huge amount of spatial data,
and their applicability for being used efficiently in various fields, including water resources
management, In more recent years, some probabilistic models such as frequency ratio (Oh et al.,
2011), multi-criteria decision analysis (MCDA) (Kaliraj et al., 2014) (Rahmati et al., 2015)
weights-of-evidence (WofE) (Pourtaghi and Pourghasemi, 2014), logistic regression (LR)
(Ozdemir, 2011b; Pourtaghi and Pourghasemi, 2014), evidential belief function (EBF) (Nampak
et al., 2014; Pourghasemi and Beheshtirad, 2015), decision tree (DT) (Chenini and Mammou,
2010), artificial neural network model (ANN) (Lee et al., 2012), and Shannon's entropy (Naghibi
et al., 2015) have been considered for groundwater potential mapping. Bivariate and multivariate
statistical models have disadvantages in measuring the relationship between groundwater
occurrence and conditioning factors (Tehrany et al., 2013; Umar et al., 2014), whereas MCDA
technique is source of bias due to expert opinion. Traditional modeling approaches are mainly
based on linear or additive modeling that is not consistent with natural process in the environment (Clapcott et al., 2013). in recent year, machine learning has proven efficient due to ability to hand
non-linear structure data from various sources with different scales. In addition, machine learning
requires no statistical assumptions. Among machine learning, ANN is considered as the most
widely used model for environmental modeling due to its computational efficiency (Bui et al.,
2016; Ghalkhani et al., 2013; Rezaeianzadeh et al., 2014). However, the ANN model has a number
of weaknesses such as poor prediction and error in modeling process (Bui et al., 2016); therefore,
hybrid models have been proposed. Among hybrid frameworks, ensemble of fuzzy logic and
Adaptive Neuro-Fuzzy Inference System (ANFIS) was reported efficient due to its high accuracy
(Güçlü and Şen, 2016; Lohani et al., 2012; Shu and Ouarda, 2008) (Chang and Tsai, 2016). It
should be noted that even though ANFIS model has a higher accuracy than the two other model
individually (Mukerji et al., 2009; Nayak et al., 2005), it has some disadvantages since it is weak
in finding the best weight parameters affecting the prediction accuracy (Bui et al., 2016). Thus,
these weights can be optimized to enhance the prediction accuracy of ground water models with
the use of machine learningoptimization algorithm.

The main aim of the current study is to carry out groundwater spring potential mapping (GSPM)
in Koohdasht-Nourabad plain, Iran using ANFIS model combined with new metaheuristic
algorithms, Invasive Weed Optimization (IWO), Differential Evolution (DE), Firefly, Particle
Swarm Optimization (PSO), and Bees algorithm (BA). Consequently, the new models have ability
to solve the weakness of the traditional ANFIS model. Another goal of the present study is drawing
a comparison between prediction capabilities of these five new hybrid models in groundwater
potential modeling in the study area as well.. Since no such studies have been published so far in
the study area, the current study is the pioneer work in this subject.

**2. Case study description**

Koohdasht-Nourabad Plain is located in the west part of the Lorestan province, Iran. It lies between
33°3′ 28 and 34° 22′ 55 N latitudes and between 46° 50′ 19 and 48° 21′ 18 E longitudes (Fig. 1).
The region is located in the semi-arid area with mean annual precipitation of about 450 mm
(Lorestan Weather Bureau report, 2016). The plain covers around 9531.9 km$^2$ with the population
of 362,000 people (according to 2016 census). The primary occupation of most people living in
the region is agriculture with groundwater is the main source. The altitude of the study area varies
between 531 m and 3175 m above the sea level, while the maximum and minimum slope is 0$^o$ and
64$^o$, respectively. Geologically, the study area is located in Zagros structural zone of Iran and is
mostly covered by Quaternary and Cretaceous-Paleocene geologic time scale. The dominant land-
use/land-cover of the study area is moderate forest (20%) and rocks covers the smallest area
percentage (0.0007%). The residential areas also covers about 3% of the Koohdasht-Nourabad
plain. Rock crop/Inceptisoils are the dominant soil types in the study area, covering about 51% of
the study area.

[Figure]

Fig.1. Groundwater well locations with DEM of the study area

**3. Methodology**

The methodological approach is shown in Fig 2..

**3.1. Data preparation**

**3.1.1. Groundwater spring inventory map**

In groundwater modeling, spatial relationship between groundwater springs and conditioning factors should be analyzed and assessed to determine the best subset of these factors. In Koohdasht-Nourabad plain, a total of 2463 springs were provided by Iranian Water Resources Management. In which, most of the spring locations were checked during extensive field surveys with GPS hand hole.

[Figure]

Fig.2. Conceptual modelling adopted in the current study

**3.1.2. Construction of the training and validation datasets**

Spatial prediction of groundwater potential mapping using machine learning model is considered
as a binary classification with two classes, spring and non-spring . Therefore, a total of 2463 non-
spring locations were randomly generated using the random point tool in ArcGIS10.2. According
to Chung and Fabbri (Chung and Fabbri, 2003), it is possible to validate the model performance
using a cross validation method that splits the dataset for the two parts. The first part is used for
building model called training dataset and the other part is utilized for validating the model
performance named as testing dataset (Pham et al., 2017a). In this study, a ratio of 70/30 was
selected randomly for generating the training and testing the dataset (Pourghasemi et al., 2013a;
Pourghasemi et al., 2012; Pourghasemi et al., 2013b; Xu et al., 2012). Accordingly, both spring
location and non-spring location have been divided into two groups for the training (1725 location)
and the validating (738 location) purposes (Fig 1).

Finally, both the training and the testing datasets were converted to raster format and then overlaid
with 13 groundwater conditioning factors to extract their attribute values, where the spring pixels
were assigned to "1" and non-spring pixels were assigned to "0" (Bui et al., 2015).

**3.1.3. Groundwater conditioning factor analysis**

**3.1.3.1. Selection of the Groundwater conditioning factor and multi-collinearity analysis**

After the initial selection of the conditioning factors, these factors should be assessed for multi-collinearity problems. Multi-collinearity takes place when two or more non-independence conditioning factors are highly correlated or in other words inter-dependent (Li et al., 2010). Several methods have been proposed to diagnose multi-collinearity, andamong them, Variance Inflation Factor (VIF) and Tolerance are widely used in environmental modeling (Bui et al., 2016; O'brien, 2007). Factors with VIF greater than 5 and tolerance less than 0.1 indicate multi-collinearity problems existed (Bui et al., 2011; O'brien, 2007).

In the current study, 13 conditioning factors have been selected including slope degree, slope aspect, altitude, plan curvature, stream power index (SPI), topographic wetness index (TWI), Terrain roughness index (TRI), distance from fault, distance from river, land-use/land-cover, rainfall, soil order, and lithology units. These factors have been determined based literature review, characteristics of the study area, and data availability (Mukherjee, 1996; Nampak et al., 2014; Oh et al., 2011; Ozdemir, 2011a). In fact, no agreement is reached on which the factors to be used for modeling. The process of converting continuous variables into categorical classes were carried out based on our frequency analysis of springs location (Khosravi et al, 2018; Ahmadisharaf et al., 2016) in order to define the class intervals (Bui et al., 2011).

Digital Elevation Model (DEM) has been downloaded from ASTER global DEM with 30x30 m grid size. Based on the DEM, slope degree, slope aspect, altitude, plan curvature, SPI, TWI and TRI were derived. Slope degree of the study areas varies between 0-64 degree. Slope factor has a direct impact on the runoff generation and groundwater recharge. As the lower the slope, the lower runoff generation and the higher groundwater recharge. The slope degree has been divided in five categories using the quantile classification scheme (Tehrany et al., 2013; Tehrany et al., 2014), including 0-5.5, 5.5-12.11, 12.11-19.4, 19.4-28.7, 28.7-64.3 degree (Fig 3a). Slope aspect is selected because it affects the groundwater potential through solar radiation. In the study area, the north aspect receives a lower sun light, and as a result, is less wet and low evapotranspiration. The slope aspect has been provided in 5 different classes including, flat, north, west, south and east (Fig 3b). The third conditioning factor is altitude. Altitude was divided into five classes using the quantile classification scheme, including 531-1070, 1070-1385, 1385-1703, 1703-2068 and 2068-3175 m (Fig.3c). Plan curvature used used with three classes, namely concave ($<-0.05$), flat ($-0.05-0.05$), and convex ($>0.05$) (Fig.3d) (Pham et al.2017). SPI is related to erosive power of surface runoff, whreas TWI links to amount of the flow that accumulates at any point in the catchment.. SPI, TWI and TRI were constructed using the Automated Geoscientific Analyses tool in SAGA-GIS 2.2 software and finally divided into five classes. They are 0-48664, 48664-227099, 227099-583969, 583969-1330153, 1330153-4136452 (Fig.3e) for SPI. For TWI, these classes are 2.1-4.6, 4.6-5.6, 5.6-6.6, 6.6-7.9, 7.9-11.9 (Fig.3f) and for TRI, these classes are 0-8.7, 8.7-18.2, 18.2-29.9, 29.9-46.6, 46.6-185 (Fig.3g).

Distance from fault and river factors have been generated using fault and river of the study area using the multiple ring-buffer tool in ArcGIS10.2. with five classes including: 0-200, 200-500, 500-1000, 1000-2000 and >2000 m (Fig. 3h and Fig. 3i). Lithology plays a key role in determining the groundwater potential occurrences due to different infiltration rate of formation that has been considered in some previous studies (Adiat et al., 2012; Nampak et al., 2014; Pradhan, 2009).

Land-use/land-cover of the study area has been provided through Landsat 7 Enhanced Thematic
Mapper plus (ETM+) images downloaded from the US Geological Survey (USGS) and supervised
image classification techniques (Lillesand et al., 2014). Finally, the accuracy of the land-use/land-
cover map has been controlled by filed surveys.

For the case of land-use/land-cover, twenty five types were recognized including agriculture,
garden, dense-forest, good rangeland, poor forest, waterway, mixture of garden and agriculture,
mixture of agriculture with dry farming, mixture of agriculture with poor-garden, dry farming,
follow, dense rangeland, very poor forest, mixture of waterway and vegetation, mixture of
moderate forest and agriculture, mixture of moderate rangeland and agriculture, mixture of poor
rangeland and follow, mixture of low forest and follow, wood-land, moderate forest, moderate
rangeland, poor rangeland, bare soil and rock, urban and residential, mixture of  very poor forest,
and rangeland have been identified and assigned to code 1 to 25 respectively (Fig.3j).

As the major source of recharge to the groundwater, rainfall has been provided via mean annual
historical rainfall data of past 15 years (2000–2015) using 4 rain-gauge stations in the study area.
Inverse distance weighted (IDW) method has been used for deriving the rainfall map with five
categories including: 300-400, 400-500, 500-600, 600-700, 700-800 mm (Fig 3k). The soil
properties directly affect the water infiltration rate as well as groundwater recharge. The 1:50,000
soil map of Lorestan province obtained from the Iranian Water Resources Department (IWRD)
has been used for the analysis. The soil map was in a polygon format which needed to be converted
to grid. The most dominant feature of the study area is rock outcrop/Entisols, rock
outcrop/Inceptisols, Inceptisols, Inceptisols/Vertisols and Badlands (Fig.3l).

[Figure]

Fig.3. Thematic Groundwater conditioning factor in the study area: slope degree(a), slope aspect (b),
altitude (c), plan curvature (d), SPI (e), TWI (f), TRI (g), distance from fault (h), distance from river (i),
land-use/land-cover (j), rainfall (k), soil order (l), and lithology units (m).

[Figure]

Fig.3.Continued

[Figure]

Fig.3. Continued

Finally, all the aforementioned groundwater conditioning factors for modeling purposes were converted to a raster grid with 30 m × 30 m in the ArcGIS 10.2 software. Lithology (unit) has a high influence on infiltration; thus, it has been considered in the current study. Lithology for the study area has been constructed in scale of 1:100000, which was provided by Iranian Department of Geology Survey (IDGS). Accordingly, thirty classes were used including: OMq, PeEf, PlQc, K1bl, Plc, pd, TRKubl, TRJvm, MPlfgp, OMql, Plbk, E2c, TRKurl, Qft2, MuPlaj, KEpd-gu, Kgu, Qft1, Ekn, KPeam, PeEtz, Kbgp, EMas-sb, Mgs, TRJlr, Klsol, JKbl, Kur, OMas and Mmn and assigned to code 1 to 30 respectively (Fig.3m).

**3.2. Spatial relationship between spring location and conditioning factors**

Step-wise Assessment Ratio Analysis (SWARA), a Multi-Criteria Decision Making (MCDM) was first introduced by Keršuliene (Keršuliene et al., 2010) was used due to both simple and rooted on experts' views SWARA has received great attention in various fields in the last five years (Alimardani et al., 2013; Hong et al., 2017). In SWARA, the expert allocates the highest and lowest rank from the most and least valuable criterion, respectively. Afterwards, the all-inclusive ranks are specified by the average value of ranks. The phases of method are as the following:

Phase one (for evolving decision making models): first, the experts define the problem solving criteria. By using the practical knowledge of the experts, the priority for each criteria are determined as well and the criteria are organized in descending order finally.

Phase two( regarding to each parameter's ranking): the following trend is employed for calculation of the weight in each criteria:

Starting from the second criterion, the respondent explains the relative importance of the criterion $j$ in relation to the $(j-1)$ criterion, and for each particular criterion as well. As Keršuliene mentioned, this process specifies the Comparative Importance of the Average Value, $S_j$ as follows (Keršuliene et al., 2010):

$$S_j = \frac{\sum_i^n A_i}{n} \tag{1}$$

where $n$ is the number of experts; $A_i$ explicates the offered ranks for each factor by the experts; j stands for the number of the factor.

Subsequently, the coefficient $K_j$ is determined as follows:

$$K_j = \begin{cases} 1 & j = 1 \\ S_j + 1 & j > 1 \end{cases} \tag{2}$$

Recalculation of weight $Q_j$ is as the following:

$$Q_j = \frac{X_{j-1}}{K_j} \tag{3}$$

The relative weights of the evaluation criteria are calculated by the following equation:

$$W_j = \frac{Q_j}{\sum_{j=1}^m Q_j} \tag{4}$$

where $W_j$ shows the relative weight of j-th criterion, and m stands for the total criteria number.

**3.3. Groundwater spring prediction modelling**

In this research, five new hybrid models namely ANFIS-DE, ANFIS-IWO, ANFIS-FA, ANFIS-PSO, ANFIS-BA were utilized for the analysis of determination of groundwater potential zonation in the study areas and for comparison between their prediction capabilities.

**3.3.1. Adaptive Neuro-Fuzzy Inference System**

Adaptive Neuro-Fuzzy Inference System (ANFIS) is obtained from the combination of Artificial Neural Network (ANN) and fuzzy logic (Jang, 1993). ANFIS has been proven more efficient than the two mentioned models in various fields (Bui et al., 2016). This is because ANN has the automatic ability but is not able to explain how to get the output from decision making. Fuzzy logic, on the other hand, is the reverse of ANN by generating output from fuzzy logical decision without the ability of self-operating learning (Aghdam et al., 2017; Chen et al., 2017b; Phootrakornchai and Jiriwibhakorn, 2015). Consequently, ANFIS was proposed to solve nonlinear and complex problems in one framework (Rezakazemi et al., 2017). This model has been used in date processing, fuzzy control and others fields (Zengqiang et al., 2008). The members of ANFIS are the function parameters from dataset for describing the system behavior (Jang, 1993). ANFIS applies to Takgi-Sugeno-Kang (TSK) fuzzy model with two rules of "If-Then" with two inputs $x_1$ and $x_2$, and one output $f$ (Takagi and Sugeno, 1985), as follows:

$$Rule2\ 1: if\ x_1\ is\ A_1\ and\ x_2\ is\ B_1, then\ f_1 = p_1 x_1 + q_1 x_2 + r_1 \tag{5}$$

$Rule\ 1: if\ x_2\ is\ A_2\ and\ x_2\ is\ B_2, then\ f_2 = p_2 x_2 + q_2 x_2 + r_2$            (6)

Jang's ANFIS consists of feed-forward neural network with six distinct layers. Detailed
description of ANFIS can be seen in (Jangs, 1993).

**3.3.2. Meta-heuristic optimization**

The main goal of this phase is to find the optimal antecedent and the consequent parameters of
the ANFIS model using IWO, DE, FA, PSO, and Bee algorithms. Fig.4 illustrates a general
methodological flow of ANFIS

[Figure]

                Fig.4. General methodological flow of ANFIS

**3.3.2.1. IWO algorithm**

[revised manuscript text omitted]
 therefore developing advanced tools for sustainable management of the groundwater is highly necessary. One of the most important tools for the management the groundwater is groundwater potential map (GPM). The current study's aim is to proposed and verified new artificial intelligence methods for spatial prediction of groundwater spring potential mapping at Koohdasht-Nourabad plain, Lorestan province, Iran. These methods are new hybrids of Adaptive Neuro-Fuzzy Inference System (ANFIS) with five meta-heuristic algorithms, Invasive Weed Optimization (IWO), Differential Evolution (DE), Firefly (FA), Particle Swarm Optimization (PSO), and Bees (BA) algorithm. Accordingly, A total of 2463 springs were identified and collected, and then divided in two subsets randomly, including 70% (1725 locations) of the total springs were used for training models, whereas the remaining 30% (738 spring locations) were utilized for the model evaluation. Thirteen groundwater conditioning factors, slope degree, slope aspect, altitude, plan curvature, stream power index (SPI), topographic wetness index (TWI), terrain roughness index (TRI), distance from fault, distance from river, land-use/land-cover, rainfall, soil order, and lithology were prepared for modeling. In the next step, the Stepwise Assessment Ratio Analysis (SWARA) method was employed to quantify the degree of relevance of these conditioning factors and the springs. The global performance of these derived models was assessed using the Area Under the curve (AUC). In addition, the Freidman and Wilcoxon signed rank test were carried out to check and confirm the best model in this study. The result showed that these models has high performance; however, the ANFIS-DE mdel has the highest prediction capability (AUC = 0.875), followed by the ANFIS-IWO model, the ANFIS-FA model (0.873), the ANFIS-PSO model (0.865), and the ANFIS-BA model (0.839). The results of this research can be useful for decision makers to sustainable management of groundwater resources.

Key words: Groundwater spring, ANFIS-DE, ANFIS-IWO, ANFIS-FA, ANFIS-PSO, ANFIS-BA, Iran.

**1. Introduction**

Groundwater is defined as the water in a saturated zone which fills rock and pore spaces (Berhanu et al., 2014; Fitts, 2002), whereas  groundwater potential is the possibility of groundwater occurrence in an area (Jha et al., 2010). The occurrence of groundwater in an aquifer is affected by various geo-environmental factors including lithology, topography, geology, fault and fracture and its connectivity, drainage pattern and land-use/land-cover (Mukherjee, 1996).  Geological strata acts like a conduit and reservoir for groundwater while. storage and transmissivity  influence  the suitability of exploitation of groundwater in a given geological formation. Downhill and depression slopes impart runoff and improve recharge and infiltration, respectively (Waikar and Nilawar, 2014).

Groundwater, which serves as a major source of drinking water to communities,  agricultural and  industrial sectors, is one of the most precious natural resources in the world (David Keith Todd and Mays, 1980) due to its consistent temperature and, widespread availability, low vulnerability to pollution, low development cost, and drought dependability (Jha et al., 2007).  Globally, 1.5 billion people are depend on  groundwater,  solely for drinking purposes, and about 38% of the irrigated lands depend on the groundwater itself (Siebert et al., 2013). Due to  population growth, , the demand  of water is constantly increased.  A major challenge now is how to have sustainable management system of groundwater to preserve and ensure continuous supply with regards to the water demand. One of the most important measures for the groundwater resource management is  to collect adequate knowledge on spatial and temporal distribution of groundwater, its quantity as well as its quality.

For the case of Iran, Approximately, two-third of the land is covered by deserts. As a result, similar to other arid regions, the main sources of water supply for drinking and  other  are the groundwater (Nosrati and Van Den Eeckhaut, 2012). Agriculture, which is one of the most prominent economic sectors in Iran, and especially, in the study area, is still be limited due to water scarcity (Zehtabian et al., 2010). Groundwater in Iran supplies around 65% of the water use-up and the remaining 35% is supplied by surface water (Rahmati et al., 2016). One of the most important measures to responsible for the increase  of fresh-water  is  to identify groundwater potential zoning,  an essential tool for performing a successful groundwater determination, protection, and management program (Ozdemir, 2011a).

There are a number of methods for groundwater exploitation in traditional approaches including drilling as well as geological, geophysical, and hydrogeological methods. Yet, they are  time-consuming,  costly  (David Keith Todd and Mays, 1980; Israil et al., 2006; Jha et al., 2010; Sander et al., 1996; Singh and Prakash, 2002). Recently, the application of geographic information systems (GIS) and remote sensing (RS) has become an effective procedure  for groundwater potential mapping (Fashae et al., 2014) due to their ability in handling huge amount of spatial data,  and their applicability for being used efficiently in various fields, including water resources management, In more recent years, some probabilistic models such as frequency ratio (Oh et al., 2011), multi-criteria decision analysis (MCDA) (Kaliraj et al., 2014) (Rahmati et al., 2015) weights-of-evidence (WofE) (Pourtaghi and Pourghasemi, 2014), logistic regression (LR) (Ozdemir, 2011b; Pourtaghi and Pourghasemi, 2014), evidential belief function (EBF) (Nampak et al., 2014; Pourghasemi and Beheshtirad,

2015), decision tree (DT) (Chenini and Mammou, 2010), artificial neural network model (ANN) (Lee et al., 2012), and Shannon's entropy (Naghibi et al., 2015) have been  considered for  groundwater potential mapping. Bivariate and multivariate statistical models have  disadvantages in measuring the relationship between groundwater occurrence and conditioning factors  (Tehrany et al., 2013; Umar et al., 2014), whereas MCDA technique is source of bias due to expert opinion. Traditional modeling approaches are  mainly based on linear or additive modeling that is not consistent with natural process in the environment (Clapcott et al., 2013).  in recent year, machine learning has proven efficient  due to ability to hand  non-linear structure  data from various  sources with different scales. In addition, machine learning requires no statistical assumptions. Among machine learning, ANN  is considered as the most widely used model for environmental modeling  due to its computational efficiency (Bui et al., 2016; Ghalkhani et al., 2013; Rezaeianzadeh et al., 2014). However, the ANN model has a number of weaknesses such as poor prediction and error in modeling process (Bui et al., 2016). therefore, hybrid models have been proposed.  Among hybrid frameworks,  ensemble of fuzzy logic  and Adaptive Neuro-Fuzzy Inference System (ANFIS) was reported efficient due to its high accuracy (Güçlü and Şen, 2016; Lohani et al., 2012; Shu and Ouarda, 2008) (Chang and Tsai, 2016). It should be noted that even though ANFIS model has a higher accuracy than the two other model individually (Mukerji et al., 2009; Nayak et al., 2005), it has some disadvantages since it is weak in finding the best weight parameters affecting the prediction accuracy (Bui et al., 2016). Thus, these weights can be optimized to enhance the prediction accuracy of ground water models with the use of machine learning optimization algorithm.

The main aim of the current study is to carry out groundwater spring potential mapping (GSPM) in Koohdasht-Nourabad plain, Iran using ANFIS model combined with new metaheuristic  algorithms,  Invasive Weed Optimization (IWO), Differential Evolution (DE), Firefly, Particle Swarm Optimization (PSO), and Bees algorithm (BA). Consequently, the new models have ability to solve the weakness of the traditional ANFIS model. Another goal of the present study is drawing a comparison between prediction capabilities of these five new hybrid models in groundwater potential modeling in the study area as well. . Since no such studies have been published so far in the study area, the current study is the pioneer work in this subject.

**2. Case study description**

Koohdasht-Nourabad Plain is located in the west part of the Lorestan province, Iran. It lies between 33°3′ 28 and 34° 22′ 55 N latitudes and between 46° 50′ 19 and 48° 21′ 18 E longitudes (Fig. 1). The region is located in the semi-arid area with mean annual precipitation of about 450 mm (Lorestan Weather Bureau report, 2016). The plain covers around 9531.9 km$^2$ with the population of 362,000 people (according to 2016 census). The primary occupation of most people living in the region is agriculture with groundwater is the main source. The altitude of the study area varies between 531 m and 3175 m above the sea level, while the maximum and minimum slope is 0° and 64°, respectively. Geologically, the study area is located in Zagros structural zone of Iran and is mostly covered by Quaternary and Cretaceous-Paleocene geologic time scale. The dominant land-use/land-cover of the study area is moderate forest (20%) and rocks covers the smallest area percentage (0.0007%). The residential areas also covers about 3% of the Koohdasht-Nourabad plain. Rock crop/Inceptisoils are the dominant soil types in the study area, covering about 51% of the study area.

[Figure]

Fig.1. Groundwater well locations with DEM of the study area

**3. Methodology**

The methodological approach is shown in Fig 2.

**3.1. Data preparation**

**3.1.1. Groundwater spring inventory map**

In  groundwater modeling, spatial relationship between
groundwater springs and  conditioning factors should be analyzed and assessed
to determine the best subset of these factors. In Koohdasht-Nourabad plain, a total of 2463 springs
were provided by  ==documentary source (==Iranian Water Resources Management) and
. In which, most of the spring locations were checked  during
extensive field survey with GPS hand hole.

[Figure]

Fig.2. Conceptual modelling adopted in the current study

**3.1.2. Construction of the training and validation datasets**

Spatial prediction of groundwater potential mapping  using machine learning model is considered
as a binary classification with two classes,
spring  and non-spring . Therefore, a total of 2463 nonspring locations were  randomly generated using the random point tool in ArcGIS10.2. According to Chung and Fabbri (Chung and Fabbri, 2003), it is possible to validate the model performance using a cross validation method that splits the dataset for the two parts . The first part is used for  building model  called training dataset and the other part is utilized for validating  the model performance named as testing dataset (Pham et al., 2017a). In this study, a ratio of 70/30 was selected randomly for generating the training and testing the dataset (Pourghasemi et al., 2013a; Pourghasemi et al., 2012; Pourghasemi et al., 2013b; Xu et al., 2012). Accordingly, both spring location and non-spring location have been divided into two groups for the training (1725 location) and the validating (738 location) purposes (Fig 1).

 Finally, both the training and the testing datasets were converted to raster format and then overlaid with 13 groundwater conditioning factors to extract their attribute values, where  the spring pixels were assigned to  "1" and non-spring pixels were assigned to "0" (Bui et al., 2015).

**3.1.3. Groundwater conditioning factor analysis**

**3.1.3.1. Selection of the Groundwater conditioning factor and multi-collinearity analysis**

After  the initial selection of the conditioning factors, the  factors should be assessed for multi-collinearity problem. Multi-collinearity takes place when two or more non-independence conditioning factors are highly correlated or in other words inter-dependent (Li et al., 2010). Several methods have been proposed to  diagnose multi-collinearity, and among them,  Variance Inflation Factor (VIF) and Tolerance are widely used  in environmental modeling  (Bui et al., 2016; O'brien, 2007). Factors with VIF greater than 5 and tolerance less than 0.1 indicate  multi-collinearity problem existed (Bui et al., 2011; O'brien, 2007).

In the current study, 13 conditioning factors have been selected including slope degree, slope aspect, altitude,  plan curvature, stream power index (SPI), topographic wetness index (TWI), Terrain roughness index (TRI), distance from fault, distance from river, land-use/land-cover, rainfall, soil order and lithology units. These factors have been determined based literature review, characteristics of the study area, and data availability (Mukherjee, 1996; Nampak et al., 2014; Oh et al., 2011; Ozdemir, 2011a). In fact, no agreement is reached  which the factors to be used for modeling. The process of converting continuous variables into categorical classes were carried out based on our frequency analysis of springs location (Khosravi et al, 2018; Ahmadisharaf et al., 2016) in order to define the class intervals (Bui et al., 2011).

Digital Elevation Model (DEM) has been downloaded from ASTER global DEM with 30x30 m grid size. Based on the DEM,  slope degree, slope aspect, altitude,  plan curvature, SPI, TWI and TRI were derived. Slope degree of the study areas varies between 0-64 degree. Slope factor has a direct impact on the runoff generation and  groundwater recharge  As the lower the slope, the lower runoff generation and the higher groundwater recharge. The slope degree has been divided in five categories using the quantile classification scheme (Tehrany et al., 2013; Tehrany et al., 2014), including 0-5.5, 5.5-12.11, 12.11-19.4, 19.4-28.7, 28.7-64.3 degree (Fig 3a). Slope aspect is selected  because it affects the groundwater potential through solar radiation. In the study area,  the north aspect receives a lower sun light, and as a result, is less wet and low evapotranspiration. The slope aspect has been provided in 5 different classes including, flat, north, west, south and east (Fig 3b). The third conditioning factor is altitude.  Altitude was divided into five classes using the quantile classification scheme, including 531-1070, 1070-1385, 1385-1703, 1703-2068 and 2068-3175 m (Fig.3c).   Plan curvature  used used with  three classes, namely concave (<−0.05), flat (−0.05–0.05), and convex (>0.05) (Fig.3d) (Pham et al.2017). SPI is related to erosive power of surface runoff, whreas  TWI  links to amount of the flow that accumulates at any point in the catchment. . SPI, TWI and TRI were constructed using the Automated Geoscientific Analyses tool in SAGA-GIS 2.2 software and finally divided into five classes. They are 0-48664, 48664-227099, 227099-583969, 583969-1330153, 1330153-4136452 (Fig.3e) for SPI. For TWI, these classes are 2.1-4.6, 4.6-5.6, 5.6-6.6, 6.6-7.9, 7.9-11.9 (Fig.3f)  and  for TRI, these classes are 0-8.7, 8.7-18.2, 18.2-29.9, 29.9-46.6, 46.6-185 (Fig.3g) .

Distance from fault and river factors have been generated using fault and river of the study area using the multiple ring-buffer command in ArcGIS10.2. with five classes including: 0-200, 200-500, 500-1000, 1000-2000 and >2000 m (Fig. 3h and Fig. 3i). Lithology plays a key role in determining the groundwater potential occurrences due to different infiltration rate of formation that has been considered in some previous studies (Adiat et al., 2012; Nampak et al., 2014; Pradhan, 2009). Land-use/land-cover of the study area has been provided through Landsat 7 Enhanced Thematic Mapper plus (ETM+) images downloaded from the US Geological Survey (USGS) and supervised image classification techniques (Lillesand et al., 2014). Finally, the accuracy of the land-use/land-cover map has been controlled by filed surveys.

 For the case of land-use/land-cover, twenty five types were recognized including agriculture, garden, dense-forest, good rangeland, poor forest, waterway, mixture of garden and agriculture, mixture of agriculture with dry farming, mixture of agriculture with poor-garden, dry farming, follow, dense rangeland, very poor forest, mixture of waterway and vegetation, mixture of moderate forest and agriculture, mixture of moderate rangeland and agriculture, mixture of poor rangeland and follow, mixture of low forest and follow, wood-land, moderate forest, moderate rangeland, poor rangeland, bare soil and rock, urban and residential, mixture of very poor forest, and rangeland have been identified and assigned to code 1 to 25 respectively  (Fig.3j).

As the major source of recharge to the groundwater, rainfall has been provided via mean annual historical rainfall data of past 15 years (2000–2015) using 4 rain-gauge stations in the study area. Inverse distance weighted (IDW) method has been used for deriving the rainfall map with five categories including: 300-400, 400-500, 500-600, 600-700, 700-800 mm
(Fig 3k). The soil properties directly affect the water infiltration rate as well as groundwater
recharge. The 1:50,000 soil map of Lorestan province obtained from the Iranian Water Resources
Department (IWRD) has been used for the analysis. The soil map was in a polygon format which
needed to be converted to grid. The most dominant feature of the study area is rock
outcrop/Entisols, rock outcrop/Inceptisols, Inceptisols, Inceptisols/Vertisols and Badlands
(Fig.3l).

[Figure]

Fig.3. Thematic Groundwater conditioning factor in the study area: slope degree(a), slope aspect (b),
altitude (c), plan curvature (d), SPI (e), TWI (f), TRI (g), distance from fault (h), distance from river (i),
land-use/land-cover (j), rainfall (k), soil order (l), and lithology units (m).

[Figure]

Fig.3.Continued

[Figure]

Fig.3. Continued

Finally, all the aforementioned groundwater conditioning factors for modeling purposes were
converted to a raster grid with 30 m × 30 m  in the ArcGIS 10.2 software. Lithology
(unit) has a high influence on infiltration; thus, it has been considered in the current study.
Lithology for the study area has been constructed in scale of 1:100000, which was  provided
by Iranian Department of Geology Survey (IDGS). Accordingly,  thirty classes
were used including: OMq, PeEf, PlQc, K1bl, Plc, pd, TRKubl, TRJvm, MPlfgp, OMql, Plbk,
E2c, TRKurl, Qft2, MuPlaj, KEpd-gu, Kgu, Qft1, Ekn, KPeam, PeEtz, Kbgp, EMas-sb, Mgs,
TRJlr, Klsol, JKbl, Kur, OMas and Mmn and assigned to code 1 to 30 respectively
(Fig.3m).

**3.2. Spatial relationship between spring location and conditioning factors**

Step-wise Assessment Ratio Analysis (SWARA),  a Multi-Criteria Decision Making (MCDM)
was first introduced by Keršuliene  (Keršuliene et al., 2010)
was used  due to both simple and rooted
on experts' views SWARA  has received  great attention in  various fields
in the last five years (Alimardani et al., 2013; Hong et al., 2017).

In SWARA,  the  expert allocates  the highest and lowest rank from the
most and least valuable criterion, respectively. Afterwards, the all-inclusive ranks are specified by
the average value of ranks. The phases of method are as the following:

Phase one  (for evolving decision making models) first, the experts define the problem solving
criteria. By using the practical knowledge of the experts, the priority for each criteria are
determined as well and the criteria are organized in descending order finally.

Phase two (regarding to each parameter's ranking) the following trend is employed for
calculation of the weight in each criteria:

Starting from the second criterion, the respondent explains the relative importance of the criterion $j$ in relation to the $(j-1)$ criterion, and for each particular criterion as well. As Keršuliene mentioned in his article, this process specifies the Comparative Importance of the Average Value, $S_j$ as follows (Keršuliene et al., 2010):

$$S_j = \frac{\sum_i^n A_i}{n} \tag{1}$$

Wwhere $n$ is the number of experts; $A_i$ explicates the offered ranks for each factor by the experts; j stands for the number of the factor.

Subsequently, the coefficient $K_j$ is determined as follows:

$$K_j = \begin{cases} 1 & j = 1 \\ S_j + 1 & j > 1 \end{cases} \tag{2}$$

Recalculation of weight $Q_j$ is as the following:

$$Q_j = \frac{X_{j-1}}{K_j} \tag{3}$$

The relative weights of the evaluation criteria are calculated by the following equation:

$$W_j = \frac{Q_j}{\sum_{j=1}^m Q_j} \tag{4}$$

wWhere $W_j$ shows the relative weight of j-th criterion, and m stands for the total criteria number.

**3.3. Groundwater spring prediction modelling**

In this research, five new hybrid models namely ANFIS-DE, ANFIS-IWO, ANFIS-FA, ANFIS-PSO, ANFIS-BA were utilized for the analysis of determination of groundwater potential zonation in the study areas and for comparison between their prediction capabilities.

**3.3.1. Adaptive Neuro-Fuzzy Inference System**

Adaptive Neuro-Fuzzy Inference System (ANFIS) is obtained from the combination of Artificial Neural Network (ANN) and fuzzy logic (Jang, 1993). ANFIS is has been proven more efficient than the two mentioned models in various fields (Bui et al., 2016). ThereforeThis is because, ANN has the automatic ability but is not able to explain how to get the output from decision making. Fuzzy logic, on the other hand, is the reverse of ANN by generating output from fuzzy logical decision without the ability of self-operating learning (Aghdam et al., 2017; Chen et al., 2017b; Phootrakornchai and Jiriwibhakorn, 2015). Consequently, ANFIS was proposed by Jang in 1993 (Jang, 1993) to solve nonlinear and complex problems in one framework (Rezakazemi et al., 2017). This model has been used in date processing, fuzzy control and others fields (Zengqiang et al., 2008). The members of ANFIS are the function parameters from dataset for describing the system behavior (Jang, 1993). ANFIS applies to Takgi-Sugeno-Kang (TSK) fuzzy model with two rules of "If-Then" with two inputs $x_1$ and $x_2$, and one output $f$ (Takagi and Sugeno, 1985), as follows:

$Rule2\ 1: if\ x_1\ is\ A_1\ and\ x_2\ is\ B_1, then\ f_1 = p_1 x_1 + q_1 x_2 + r_1$ \hfill (5)

$Rule\ 1: if\ x_2\ is\ A_2\ and\ x_2\ is\ B_2, then\ f_2 = p_2 x_2 + q_2 x_2 + r_2$ \hfill (6)

Jang's ANFIS consists of feed-forward neural network with six distinct layers. Detailed
description of ANFIS model described in details atcan be seen in (Jangs, 1993).

**3.3.2. Meta-heuristic optimization**

The main goal of this phase is to find the optimal antecedent and the consequent parameters of
the ANFIS model using IWO, DE, FA, PSO, and Bee algorithms. Fig.4 illustrates a general
methodological flow of ANFIS The processing in MATLAB software is shown in Fig.4.

[Figure]

Fig.4. General methodological flow of ANFISprocessing of ANFIS hybrid model

**3.3.2.1. IWO algorithm**

[revised manuscript text omitted]

---

## Author Comment (AC2) · 6 Apr 2018

Thank you so much for your positive and valuable comments. This document explains the changes made in the revised manuscript while dealing with the comments raised

by the reviewers.

Comment 1: There are many grammatical mistakes appeared in the article. I strongly request that this manuscript should be totally edited by professional English editors for improving the English writing.

Answer: We have carefully checked and revised English in the manuscript.

Comment 2: In Figure 2, thirteen groundwater conditioning factors were served as input of hybrid models. The sensitivity analysis should be performed to investigate that which conditioning factor is most important factor to affect the output.

Answer: Thanks for your valuable suggestion. It was performed using one of the most widely used methods namely Information Gain Ration (IGR) as follows:

4.2. Determination of the most important parameters The most common method of information gain ratio (IGR) was applied to identification of the most important conditioning factors. Result shows that all thirteen conditioning factors are effective on groundwater occurrences as the land-use/landcover factor has the most important impact on groundwater (IGR=0.502) followed by lithology (IGR=0.465), rainfall (IGR=0.421), TWI (IGR=0.400), soil (IGR=0.370), TRI (IGR=0.337), slope degree (IGR=0.317), altitude (IGR=0.287), distance to river (IGR=0.139), aspect (IGR=0.066), plan curvature (IGR=0.0548), distance to fault (IGR=0.0482) and SPI (IGR=0.0323).

Comment 3: How many non-spring stations in the study area? Why are the numbers of non-spring stations should same with spring stations?

Answer: In the present study the same number of 2463 springs and non-springs location have been considered which stated in the paper. Number of non-springs location can be different from spring location (higher or less than), but it is better be the same, as the results of the models depend on them and also with testing dataset (30% of both of spring and non-springs location) the prediction ability of the model is evaluated, thus, for example, if there is a lot of non-spring location in compare to spring location,

and may be this location located on the non-spring potential area, thus accuracy of the model would be increased in inaccuracy method. But for achieve the better and more accurate result it is better that has a same number.

Comment 4: In line 171, "In the current study, 14 conditioning factors.." should be "13 conditioning factors.."

Answer: Thank you for your precis attention; it was corrected.

Comment 5: I am wondering that Figures 4, 5, and 6 are necessary, because these figures are taken from other literature.

Answer: Thanks for your valuable comment, the authors agree with you and removed these three Figures from the paper.

Comment 6: In equation (1), what is "I"? The term "I" should be "i=1".

Answer: "I" is the index of each expert. We must write SWARA method completely in method. But, we added number of expert for determining the priority for each criterion.

Comment 7: Pages 12, 13, and 14, the authors spend many spaces to describe the algorithm of ANFIS model. Actually, we can find the same description in many references. I am also wondering that the description of ANFIS model is necessary. Should it move to Appendix.

Answer: Thanks for this valuable comment. The extra explanation has been removed and proper reference added for more details for ANFIS model.

Comment 8: In "Discussion" Section, lines 708-719, why are these sentences put here? These sentences seem to repeat again.

Answer: Thank you for your precis attention and valuable comment; it was removed from the paper as they are repeated.

Comment 9: Can the authors describe that how much time you spend to run for each

hybrid ANFIS model in MATLAB environment?

Answer: Yes, we have proved a figure that show the processing time for each model in the revised manuscript (Fig.7).

Comment 10. The important factors to be adjusted for each hybrid ANFIS model should be listed with a Table.

Answer: Processing in Matlab shown in Figure 4. FIS is structure and many variants. Consequently, show all optimize parameters for each hybrid models are very difficult and get many space in paper if presented in table format. So, our suggestion is replacement of table of optimize parameters using figure 7 for describe processing. This Figure was added to the paper as well.

Dear Editor and reviewers: Thank you so much for your viewpoints and comments in regarding our manuscript. I hope the emendations caused to consent the respected reviewers and editor-in-chief and made my paper well qualified for publication.

Please also note the supplement to this comment:
https://www.hydrol-earth-syst-sci-discuss.net/hess-2017-707/hess-2017-707-AC2-supplement.pdf

---

## Editor Comment (EC1) · D. Solomatine (Editor) · 7 Apr 2018

Dear authors,

[Figure]

You will be invited revise the paper, and it will be sent to the referees. In your replies, please provide answers to all comments (most will be the same as you have already given in Interactive Discussion), explain what do you change in the manuscript, and clearly indicate (e.g. in color) what have you changed in the new version of the manuscript.

Please take into account the referees' comments, and also the following: - give serious attention to improving English. - please consider shortening the paper. It is really very long. Consider removing text-book material (provide references instead). Clearly describe what you have done, and your results - make your message shorter. It is often useful to make Apendices and to put additional material there.

Good luck.

---

## Author Response (AR1)

**List of changes in the revised paper:**

This document explains the changes made in the revised manuscript while dealing with the comments raised by the reviewers. Editor's and reviewer's comments are marked in **black** colour while author's response is shown in **Red** text.

**Editor's comment:**

**Comment 1:** please shorten the title, it is really too long. There is no need to mention all methods.

Answer:

Thanks for your valuable comment. It was shortened as follows:

Spatial Prediction of Groundwater Spring Potential Mapping Based on Adaptive Neuro-Fuzzy Inference System and Metaheuristic Optimization

**Comment 2:** In your replies, please provide answers to all comments (most will be the same as you have already given in Interactive Discussion), explain what do you change in the manuscript, and clearly indicate (e.g. in color) what have you changed in the new version of the manuscript.

Answer: All changes were highlighted in green.

**Comment 3:** give serious attention to improving English.

Answer:

An English language was corrected seriously, which for the last time was corrected using Prof. Bjørn Kristofersen at University of South-Eastern Norway.

**Comment 4:** Please consider shortening the paper. It is really very long. Consider removing text-book material (provide references instead)

Answer:

Thanks for your suggestion and valuable comment. The paper shortened as far as we could, from 14,800 words to 10,900 words, and proper citation has been added for more details.

Comment 5: Clearly describe what you have done, and your results - make your message

shorter. It is often useful to make Apendices and to put additional material there.

Answer:

Thanks for your valuable comment. The results section has been corrected as follows:

Groundwater is the most important natural resource in the world and about 25 percent of all fresh water is estimated as groundwater. Thus, the groundwater potential mapping has been considered as one of the most effective methods for the management of groundwater resources for better exploitation. The main result of the present study can be summarized as:

- 1- The results showed that although all models had very good reasonable results, but, the ANFIS-DE had the highest prediction power (0.875) followed by ANFIS-IWO and ANFIS-FA (0.873), ANFIS-PSO (0.865) and ANFIS-BA (0.839).
- 2- According to the results of the SWARA method, most springs existed in an altitude of 1703-2068 m, flat curvature, east aspect, TWI of 6.6-7.9, TRI of 0-8.7, SPI of 583969-1330153, Inceptisols soil, slope of 0-5.5 degree, 0-200 m distance from river, 500-1000 m distance from fault, rainfall between 500-600 mm, in a garden, in a Pliocene-Quaternary lithological age and OMq lithology unit at the case study.
- 3- Based on the information gain ratio, the most important factors on the groundwater occurrence are land-use/land-cover, lithology, rainfall and TWI, but the least important factors are plan curvature, distance to fault and SPI.
- 4- Based on the ANFIS-DE model, totally 39.33% of the case study have a high and very high groundwater potential placed at north of the case study.

**Dear Editor: Thank you so much for your positive and valuable comments.**

**Reviewer 1:**

**Comment 1**: I think title of the mentioned research is very long; please authors try to decrease it.

Answer:

Thanks for you valuable comment. Authors agree with you, the title have been shorted as follows:

Spatial Prediction of Groundwater Spring Potential Mapping Based on Adaptive Neuro-Fuzzy Inference System and Metaheuristic Optimization

**Comment 2**: In abstract, your means from curvature is which one? Plan or profile? Answer:

Our mean is plan curvature which was corrected from throughout the paper.

**Comment 3**: In abstract, what is your means from soil order? Answer:

To identify, understand, and manage soils, soil scientists have developed a soil classification or taxonomy system. The most general level of classification in the United States system is the soil order, of which there are 12 (such as Alfisols, Aridisoils, and etc.). Each order is based on one or two dominant physical, chemical, or biological properties that differentiate it clearly from the other orders.

**Comment 4**: Results of models are very similar together. Please edit results of lines 33-35.

Answer:

Thanks for this valuable comment. The sentences have been corrected as:

The result showed that all models have high performance; however, the ANFIS-DE model has the highest prediction capability (AUC = 0.875), followed by the ANFIS-IWO model, the ANFIS-FA model (0.873), the ANFIS-PSO model (0.865), and the ANFIS-BA model (0.839).

**Comment 5**: Please add a reference in lines 118-119 for rainfall descriptions. Answer:

The proper references have been added as 'Iran Meteorological Organization''

Comment 6: Quality of Fig. 2 isn't proper. Please draw it again.

Answer:

This Figure was draw and corrected again and added to the paper.

**Comment 7**: Please add source of groundwater spring inventory map Answer:

The proper source have been added: a total of 2463 springs were selected from documentary source (Iranian Water Resources Management) and considered for modeling.

**Comment 8**: Please explain about classification of different layers or at least add some citations for the mentioned classifications.

Answer:

Some references have been added to then sentences as:

The process of converting continuous variables into categorical classes were carried out based on our frequency analysis of springs location (Khosravi et al, 2018) in order to define the class intervals

**Comment 9**: Fig. 3 (j) and 3 (m) what are codes?

Answer:

Thanks for your punctuality, it was corrected and considered at the paper. The new figures for them were draw.

**Comment 10**: According to Table 2, I think it isn't a land use map, it is land cover. Please change its name or present land use/land cover Answer: Thanks for your punctuality; it was corrected to land-use/land-cover throughout the paper.

Dear Reviewer 1: Thank you so much for your positive and valuable comments.

**Reviewer 2:**

**Comment 1**: There are many grammatical mistakes appeared in the article. I strongly request that this manuscript should be totally edited by professional English editors for improving the English writing. Answer:

We have carefully checked and revised English in the manuscript.

**Comment 2**: In Figure 2, thirteen groundwater conditioning factors were served as input of hybrid models. The sensitivity analysis should be performed to investigate that which conditioning factor is most important factor to affect the output.

**Answer:**

Thanks for your valuable suggestion. It was performed using one of the most widely used methods namely Information Gain Ration (IGR) as follows:

4.2. Determination of the most important parameters

The most common method of information gain ratio (IGR) was applied to identification of the most important conditioning factors. Result shows that all thirteen conditioning factors are effective on groundwater occurrences as the land-use/landcover factor has the most important impact on groundwater (IGR=0.502) followed by lithology (IGR=0.465), rainfall (IGR=0.421), TWI (IGR=0.400), soil (IGR=0.370), TRI (IGR=0.337), slope degree (IGR=0.317), altitude (IGR=0.287), distance to river (IGR=0.139), aspect (IGR=0.066), plan curvature (IGR=0.0548), distance to fault (IGR=0.0482) and SPI (IGR=0.0323).

**Comment 3**: How many non-spring stations in the study area? Why are the numbers of non-spring stations should same with spring stations? Answer:

In the present study the same number of 2463 springs and non-springs location have been considered which stated in the paper. Number of non-springs location can be different from spring location (higher or less than), but it is better be the same, as the result of the models depends on it and also with testing dataset (30% of both of spring and non-springs location) the prediction ability of the model is evaluated, thus, for example, if there is a lot of non-spring location in compare to spring location, and may be this

location located on the non-spring potential area, thus accuracy of the model would be increased in inaccuracy method. But for achieve the better and more accurate result it is better that has a same number.

**Comment 4**: In line 171, "In the current study, 14 conditioning factors." should be "13 conditioning factors."

Answer:

Thanks for your punctuality, it was corrected.

**Comment 5**: I am wondering that Figures 4, 5, and 6 are necessary, because these figures are taken from other literature.

Answer:

Thanks for your valuable comment, the authors agree with you and removed these three Figures from the paper.

**Comment 6**: In equation (1), what is "I"? The term "I" should be "i=1".

Answer:

"I" is the index of each expert. We must write SWARA method completely in method. But, we added number of expert for determining the priority for each criterion.

**Comment 7**: Pages 12, 13, and 14, the authors spend many spaces to describe the algorithm of ANFIS model. Actually, we can find the same description in many references. I am also wondering that the description of ANFIS model is necessary. Should it move to Appendix.

Answer:

Thanks for this valuable comment. The extra explanation has been removed and proper reference added for more details for ANFIS model.

**Comment 8**: In "Discussion" Section, lines 708-719, why are these sentences put here? These sentences seem to repeat again.

Answer:

Thanks for your punctuality and valuable comment; it was removed from the paper as they are repeated.

**Comment 9**: Can the authors describe that how much time you spend to run for each hybrid ANFIS model in MATLAB environment?

Answer:

Yes, we have proved a figure that show the processing time for each model in the revised manuscript (Fig.1).

Fig. 1. Processing time used for training the models

**Comment 10**. The important factors to be adjusted for each hybrid ANFIS model should be listed with a Table.

**Answer:**

Processing in Matlab shown in Figure 2. FIS is structure and many variants. Consequently, show all optimize parameters for each hybrid models are very difficult and get many space in paper if presented in table format. So, our suggestion is replacement of table of optimize parameters using figure 1 for describe processing. This Figure was added to the paper as well.

Fig.2. processing of ANFIS hybrid model

---

## Author Response (AR2)

Dear Dr.Dimitri Solomatine:

Thank you so much for your viewpoints and comments in regarding our manuscript. Also, thank you for accepting and publishing this research in HESS.

**Editor's comment:**

L 250: Takgy --> Takagi

Lines 256, 284, 302: "can be seen" -- better to say "can be found"

- Thank you and I agree with you. I change them in manuscript.

When submitting, please ensure the new version of the title is specified. Spatial Prediction of Groundwater Spring Potential Mapping Based on Adaptive Neuro-Fuzzy Inference System and Metaheuristic Optimization (so that there is no "1" in the middle)

- I am so sorry. I eliminate "1".